# Neural Circuit Diagrams: Robust Diagrams for the Communication, Implementation, and Analysis of Deep Learning Architectures

**Vincent Abbott**  *Vincent.Abbott@anu.edu.au*
*Australian National University*

**Reviewed on OpenReview:** *https://openreview.net/forum?id=RyZB4qXEgt*

## Abstract

Diagrams matter. Unfortunately, the deep learning community has no standard method for diagramming architectures. The current combination of linear algebra notation and ad-hoc diagrams fails to offer the necessary precision to understand architectures in all their detail. However, this detail is critical for faithful implementation, mathematical analysis, further innovation, and ethical assurances. I present neural circuit diagrams, a graphical language tailored to the needs of communicating deep learning architectures. Neural circuit diagrams naturally keep track of the changing arrangement of data, precisely show how operations are broadcast over axes, and display the critical parallel behavior of linear operations. A lingering issue with existing diagramming methods is the inability to simultaneously express the detail of axes and the free arrangement of data, which neural circuit diagrams solve. Their compositional structure is analogous to code, creating a close correspondence between diagrams and implementation.

In this work, I introduce neural circuit diagrams for an audience of machine learning researchers. After introducing neural circuit diagrams, I cover a host of architectures to show their utility and breed familiarity. This includes the transformer architecture, convolution (and its difficult-to-explain extensions), residual networks, the U-Net, and the vision transformer. I include a Jupyter notebook that provides evidence for the close correspondence between diagrams and code. Finally, I examine backpropagation using neural circuit diagrams. I show their utility in providing mathematical insight and analyzing algorithms' time and space complexities.

## 1 Introduction

### 1.1 Necessity of Improved Communication in Deep Learning

Deep learning models are immense statistical engines. They rely on components connected in intricate ways to slowly nudge input data toward some target. Deep learning models convert big data into usable predictions, forming the core of many AI systems. The design of a model—its architecture—can significantly impact performance (Krizhevsky et al., 2012), ease of training (He et al., 2015; Srivastava et al., 2015), generalization (Ioffe & Szegedy, 2015; Ba et al., 2016), and ability to efficiently tackle certain classes of data (Vaswani et al., 2017; Ho et al., 2020).

Architectures can have subtle impacts, such as different image models recognizing patterns at various scales (Ronneberger et al., 2015; Luo et al., 2017). Many significant innovations in deep learning have resulted from architecture design, often from frighteningly simple modifications (He et al., 2015). Furthermore, architecture design is in constant flux. New developments constantly improve on state-of-the-art methods (He et al., 2016; Lee, 2023), often showing that the most common designs are just one of many approaches worth investigating (Liu et al., 2021; Sun et al., 2023).

However, these critical innovations are presented using ad-hoc diagrams and linear algebra notation (Vaswani et al., 2017; Goodfellow et al., 2016). These methods are ill-equipped for the non-linear operations and actions on multi-axis tensors that constitute deep learning models (Xu et al., 2023; Chiang et al., 2023). Furthermore, these tools are insufficient for papers to present their models in full detail. Subtle details such as the order of normalization or activation components can be missing, despite their impact on performance (He et al., 2016).

Works with immense theoretical contributions can fail to communicate equally insightful architectural developments (Rombach et al., 2022; Nichol & Dhariwal, 2021). Many papers cannot be reproduced without reference to the accompanying code. This was quantified by Raff (2019), where only 63.5% of 255 machine learning papers from 1984 to 2017 could be independently reproduced without reference to the author's code. Interestingly, the number of equations present was *negatively* correlated with reproduction, further highlighting the deficits of how models are currently communicated. The year that papers were published had no correlation to reproducibility, indicating that this problem is not resolving on its own.

Relying on code raises many issues. The reader must understand a specific programming framework, and there is a burden to dissect and reimplement the code if frameworks mismatch. Without reference to a blueprint, mistakes in code cannot be cross-checked. The overall structure of algorithms is obfuscated, raising ethical risks about how data is managed (Kapoor & Narayanan, 2022).

Furthermore, papers that clearly explain their models without resorting to code provide stronger scientific insight. As argued by Drummond (2009), replicating the code associated with experiments leads to weaker scientific results than reproducing a procedure. After all, replicating an experiment perfectly controls *all* variables, including irrelevant ones, making it difficult to link any independent variable to the observed outcome.

However, in machine learning, papers often cannot be independently reproduced without referencing their accompanying code. As a result, the machine learning community misses out on experiments that provide general insight independent of specific implementations. Improved communication of architectures, therefore, will offer clear scientific value.

## 1.2 Case Study: Shortfalls of *Attention is All You Need*

To highlight the problem of insufficient communication of deep learning architectures, I present a case study of *Attention is All You Need*, the paper that introduced transformer models (Vaswani et al., 2017). Introduced in 2017, transformer models have revolutionized machine learning, finding applications in natural language processing, image processing, and generative tasks (Phuong & Hutter, 2022; Lin et al., 2021).

Transformers' effectiveness stems partly from their ability to inject external data of arbitrary width into base data. I refer to axes representing the number of items in data as a **width**, and axes indicating information per item as a **depth**.

An **attention head** gives a weighted sum of the injected data's value vectors, $V$. The weights depend on the attention score the base data's query vectors, $Q$, assign to each key vector, $K$, of the injected data. Value and key vectors come in pairs. Fully connected layers, consisting of learned matrix multiplication, generate $Q$, $K$, and $V$ vectors from the original base and injected data. **Multi-head attention** uses multiple attention heads in parallel, enabling efficient parallel operations and the simultaneous learning of distinct attributes.

*Attention is All You Need*, which I refer to as the original transformer paper, explains these algorithms using diagrams (see Figure 1) and equations (see Equation 1,2,3) that hinder understandability (Chiang et al., 2023; Phuong & Hutter, 2022).

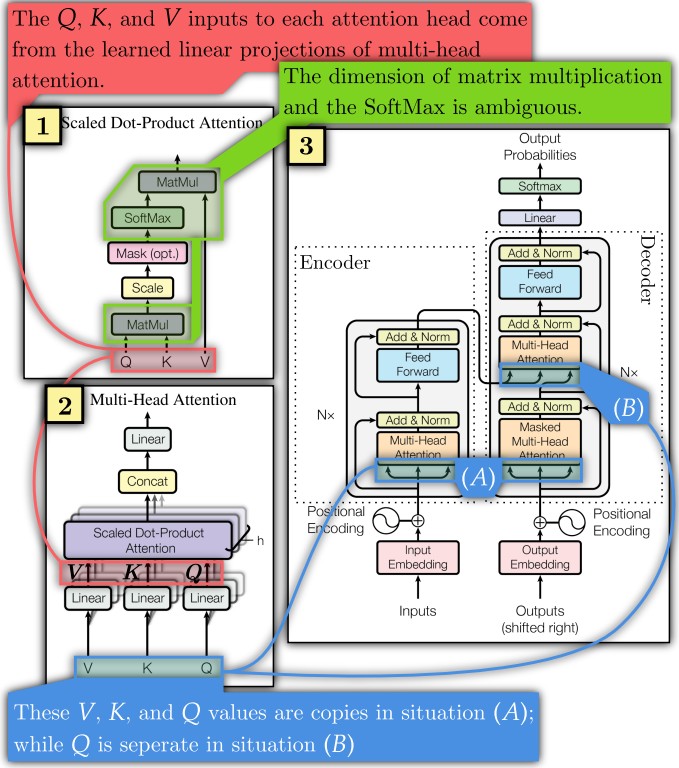

Figure 1: My annotations of the diagrams of the original transformer model. Critical information is missing regarding the origin of $Q$, $K$, and $V$ values (**red** and **blue**), and the axes over which operations act (**green**).

$$\text{Attention}(Q, K, V) = \text{SoftMax}\left(\frac{QK^T}{\sqrt{d_k}}\right) V \ (d_k \text{ is the key depth}) \tag{1}$$

$$\text{MultiHead}(Q, K, V) = \text{Concat}(\text{head}_1, ..., \text{head}_h)W^O \tag{2}$$

$$\text{where head}_i = \text{Attention}\left(QW_i^Q, KW_i^K, VW_i^V\right) \tag{3}$$

The original transformer paper obscures dimension sizes and their interactions. The dimensions over which SoftMax[1] and matrix multiplication operates is ambiguous (Figure 1.1, **green**; Equation 1, 2, 3).

Determining the initial and final matrix dimensions is left to the reader. This obscures key facts required to understand transformers. For instance, $K$ and $V$ can have a different width to $Q$, allowing them to inject external information of arbitrary width. This fact is not made clear in the original diagrams or equations. Yet, it is necessary to understand why transformers are so effective at tasks with variable input widths, such as language processing.

The original transformer paper also has uncertainty regarding $Q$, $K$, and $V$. In Figure 1.1 and Equation 1, they represent separate values fed to each attention head. In Figure 1.2 and Equation 2 and 3, they are all copies of each other at location *(A)* of the overall model in Figure 1.3, while $Q$ is separate in situation *(B)*.

Annotating makeshift diagrams does not resolve the issue of low interpretability. As they are constructed for a specific purpose by their author, they carry the author's curse of knowledge (Pinker, 2014; Hayes & Bajzek, 2008; Ross et al., 1977). In Figure 1, low interpretability arises from missing critical information, not from insufficiently annotating the information present. The information about which axes are matrix multiplied or are operated on with the SoftMax is simply not present.

---

[1]Using $i$ and $k$ to index over data, we have $\text{SoftMax}(\mathbf{v})[i] = \exp(\mathbf{v}[i])/\Sigma_k \exp(\mathbf{v}[k])$.

Therefore, we need to develop a framework for diagramming architectures that ensures key information, such as the axes over which operations occur, is automatically shown. Taking full advantage of annotating the critical information already present in neural circuit diagrams, I present alternative diagrams in Figures 20, 21, and 22.

### 1.3 Current Approaches and Related Works

These issues with the current ad-hoc approaches to communicating architectures have been identified in prior works, which have proposed their own solutions (Phuong & Hutter, 2022; Chiang et al., 2023; Xu et al., 2023; Xu & Maruyama, 2022). This shows that this is a known issue of interest to the deep learning community. Non-graphical approaches focus on enumerating all the variables and operations explicitly, whether by extending linear algebra notation (Chiang et al., 2023) or explicitly describing every step with pseudocode (Phuong & Hutter, 2022). Visualization, however, is essential to human comprehension (Pinker, 2014; Borkin et al., 2016; Sadoski, 1993). Standard non-graphical methods are essential to pursue, and the community will benefit significantly from their adoption; however, a standardized graphical language is still needed.

The inclination towards visualizing complex systems has led to many tools being developed for industrial applications. Labview, MATLAB's Simulink, and Modelica are used in academia and industry to model various systems. For deep learning, TensorBoard and Torchview have become convenient ways to graph architectures. These tools, however, do not offer sufficient detail to implement architectures. They are often dedicated to one programming language or framework, meaning they cannot serve as a general means of communicating new developments. Besides, a rigorously developed framework-independent graphical language for deep learning architectures would help to improve these tools. This requires diagrams equipped with a mathematical framework that captures the changing structure of data, along with key operations such as broadcasting and linear transformations.

Many mathematically rigorous graphical methods exist for a variety of fields. This includes Petri nets, which have been used to model several processes in research and industry (Murata, 1989). Tensor networks were developed for quantum physics and have been successfully extended to deep learning (Biamonte & Bergholm, 2017; Xu et al., 2023; Xu & Maruyama, 2022). Xu et al. (2023) showed that re-implementing models after making them graphically explicit can improve performance by letting parallelized tensor algorithms be employed. Robust diagrams, therefore, can benefit both the communication and performance of architectures. Formal graphical methods have also been developed in physics, logic, and topology (Baez & Stay, 2010; Awodey, 2010).

All these graphical methods have been found to represent an underlying category, a mathematical space with well-defined composition rules (Meseguer & Montanari, 1990; Baez & Stay, 2010). A category theory approach allows a common structure, monoidal products, to define an intuitive graphical language (Selinger, 2009; Fong & Spivak, 2019). Category theory, therefore, provides a robust framework to understand and develop new graphical methods.

However, a noted issue (Chiang et al., 2023) of previous graphical approaches is they have difficulty expressing non-linear operations. This arises from a *tensor approach* to monoidal products. Data brought together cannot necessarily be copied or deleted. This represents, for instance, axes brought together to form a matrix and this approach makes linear operations elegantly manageable. It, however, makes expressing copying and deletion impossible. The alternative Cartesian approach allows copying and deletion, reflecting the mechanics of classical computing.

The Cartesian approach has been used to develop a mathematical understanding of deep learning (Shiebler et al., 2021; Fong et al., 2019; Wilson & Zanasi, 2022; Cruttwell et al., 2022). However, Cartesian monoidal products do not automatically keep track of dimensionality and cannot easily represent broadcasting or linear operations. These works often rely on the most rudimentary model of deep-learning networks as sequential linear layers and activation functions, despite residual networks having become the norm (He et al., 2015; 2016). The graphical language generated by a pure Cartesian approach fails to show the details of architectures, limiting its ability to consider models as they appear in practice.

The issue of only looking at rudimentary, linear layer-activation layer models is pervasive in deep learning research (Zhang et al., 2017; Saxe et al., 2019; Li et al., 2022). There are uncountably many ways of relating inputs to outputs. Every theory or hypothesis about deep learning algorithms has to assume that we are working with some subset of all possible functions. However, specifying this subset means theoretical insights can only apply to that subset. This precludes us from using such theories to compare disparate architectures and make design choices.

The problem of only considering rudimentary models is partially a consequence of us not having the tools to robustly represent more complex models, never mind the tools to confidently analyze them. Category theory-based diagrams can serve as models of intricate systems. Structure-preserving maps allow analyses to scale over entire models. Therefore, developing comprehensive diagrams that correspond to mathematical expressions can be the first step in a rigorous theory of deep learning architectures with clear practical applications.

The literature reveals a combination of problems that need to be solved. Deep learning suffers from poor communication and needs a graphical language to understand and analyze architectures. Category theory can provide a rigorous graphical language but typically forces a choice between tensor or Cartesian approaches. The elegance of tensor products and the flexibility of Cartesian products must both be available to properly represent architectures. A category arises when a system has sufficient compositional structure, meaning a non-category theory approach to diagramming architectures will likely yield a category anyway. The challenge of reconciling Cartesian and tensor approaches, therefore, remains.

### 1.4 The Philosophy of My Approach

As I am introducing these diagrams, I have a burden to explain how I think they should be used and to address criticisms of creating a diagramming standard in the first place. I will take a brief aside to address these points, which I believe will aid in the adoption of neural circuit diagrams.

These diagrams are intended to express sequential-tensor deep learning models. This is in contrast to machine learning or artificial intelligence systems more generally. Deep learning models are machine learning models with sequential data processing through neural network layers. I do not cover recursive or branching models in this work. Furthermore, I assume data is always in the form of tuples of tensors. Generalizing diagrams to further contexts is an exciting avenue for future research.

By making these assumptions, I develop diagrams specialized for some of the most essential but difficult-to-explain systems in artificial intelligence research. Researchers outside the narrow scope of sequential-tensor deep learning models often rely on these tools. By more clearly communicating them, researchers who may not be up to date on the latest innovations or aware of their options stand to benefit an immense deal.

I do not expect two independent teams to diagram architectures the exact same way. Indeed, I do not believe the appropriate diagramming framework would have this property. Diagrams should have the flexibility to allow for innovations and to appeal to the audience's level of knowledge. Instead, the benefit of my framework is to have comprehensive, robust diagrams with clear correspondence to implementation and analysis, in contrast to ad-hoc diagrams, which often fail to include critical information.

Neural circuit diagrams can be decomposed into sections that allow for layered abstraction. The exact details of code can be abstracted into single-symbol components. Sections of diagrams can be highlighted for the reader's clarity, and repeated patterns can be defined as components. Diagrams have an immense compositional structure. The horizontal axis represents sequential composition, and the vertical axis represents parallel composition. Sections and components can be joined like Lego bricks to construct models.

This sectioning allows for a close correspondence between diagrams and implementation. Every highlighted section becomes a module in code. Diagrams, therefore, provide a cross-platform blueprint for architectures. This allows implementations to be cross-checked to a reference, increasing reliability. Furthermore, which components are abstracted and the level of abstraction can vary depending on the audience, leading to clearer, specialized communication.

A common criticism is that introducing a new standard simply increases the number of standards, worsening the issue trying to be solved (below). I do not believe this is a relevant critique for deep learning diagrams. Currently, there are no standard diagramming methods. Every paper, in a sense, has its own ad-hoc diagramming scheme. Compared to this, neural circuit diagrams only need to be learned once, after which architectures can be clearly and explicitly explained. Furthermore, they build on existing research on robust monoidal string diagrams, which have been found to be a universal standard for various fields (Baez & Stay, 2010).

### 1.5 Contributions

To address the need for more robust communication and analysis of deep learning architectures, I introduce neural circuit diagrams. Neural circuit diagrams solve the lingering challenge of accommodating both the details of axes (the tensor approach) and the free arrangement of data (the Cartesian approach) in diagrams. They are specialized for sequential algorithms on memory states consisting of tuples of tensors.

Diagramming the details of axes means the shape of data is clear throughout a model. They easily show broadcasting and provide a graphical calculus to rearrange linear functions into equivalent forms. At the same time, they clearly represent tuples, copying, and deletion, processes that typical graphical methods struggle with. This makes them uniquely capable of accurately representing deep learning models.

Inspired by category theory and especially monoidal string diagrams (Selinger, 2009; Baez & Stay, 2010), this work builds on a literature of robust diagramming methods. However, the category theory details are omitted to maximize impact among machine learning researchers.

The benefits of neural circuit diagrams are many. They allow for clearer communication of new developments, making ideas more rapidly disseminated and understood. They offer robust blueprints for designing and implementing models, accelerating innovation and streamlining productivity. Furthermore, they allow for rigorous mathematical analysis of architectures, bringing us closer to a theoretical understanding of deep learning.

These points are evidenced by diagramming a host of architectures. I cover a basic multi-layer perceptron, the transformer architecture, convolution (and its difficult-to-explain permutations), the identity ResNet, the U-Net, and the vision transformer. I provide a Jupyter notebook that implements these diagrams, which provides further evidence for the close relationship between diagrams and implementation. Finally, I offer a novel analysis of backpropagation, which shows the utility of neural circuit diagrams for rigorous analysis of architectures.

## 2 Reading Neural Circuit Diagrams

### 2.1 Commutative Diagrams

We aim to craft diagrams that precisely represent deep learning algorithms. While these diagrams will eventually be generalized, we will initially concentrate on common models. Specifically, we will explore models that successively process data of predictable types. To facilitate understanding, we will introduce diagrams of gradually increasing complexity. To begin, let's delve into an intuitive diagram, where symbols represent data types, and arrows signify the functions connecting them.

Note, I use forward composition with ";", meaning $f : \mathrm{str} \rightarrow \mathrm{int}$ composes with $g : \mathrm{int} \rightarrow \mathrm{float}$ by $(f; g) : \mathrm{str} \rightarrow \mathrm{float}$.

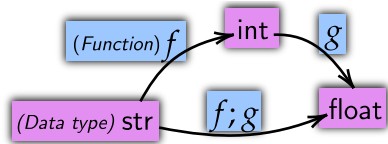

Figure 2: We have two functions: $f : \mathrm{str} \rightarrow \mathrm{int}$ and $g : \mathrm{int} \rightarrow \mathrm{float}$. These functions can be composed into a single function $(f; g) : \mathrm{str} \rightarrow \mathrm{float}$. In commuting diagrams, we represent data types, such as str, int, and float, with floating symbols, while functions are denoted by arrows connecting them.

### 2.1.1 Tuples and Memory

Algorithms are rarely composed of operations on a single variable. Instead, their steps involve operations on memory states composed of multiple variables. The data type of a memory state is a tuple of the variables which compose it. So, a state containing an int and a str would have a type int × str.

Consider a single algorithmic step acting on a compound memory state $A \times B \times C$. A function $f : B \times C \to D$ acting on this memory state would give an overall step with shape $\text{Id}[A] \times f : A \times (B \times C) \to A \times D$. Note that $\text{Id}[A]$ is the identity. We need to indicate $A$, even though $f$ does not act on it, so that the initial and final memory states are properly shown. In Figure 3, I diagram $f$ along another function $g : A \times D \to E$.

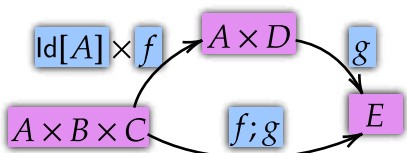

Figure 3: Here, I diagram two functions, $f : B \times C \to D$ and $g : A \times D \to E$, acting together. To represent the full memory states, we are required to amend $f$ into $\text{Id}[A] \times f : A \times (B \times C) \to A \times D$. The composed function is $(\text{Id}[A] \times f); g : A \times (B \times C) \to E$.

## 2.2 String Diagrams

These commuting diagrams fall short, however. As algorithms scale, operations and memory states get more complex. Usually, functions only act on some variables. However, it is not clear how to these targeted functions. Compound data types and compound functions are better suited by reorienting diagrams as in Figure 4. We will have horizontal wires represent types, and symbols represent functions. Diagrams are forced to horizontally go left to right.

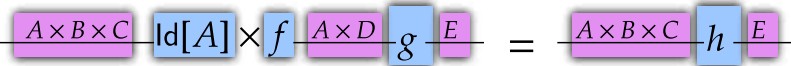

Figure 4: We reorient diagrams to go left to right. Wires represent data types, and symbols represent functions. This expression defines $h$.

This reorientation allows us to represent compound types and functions easily. We can diagram tupled types $A \times B$ as a wire for $A$ and a wire for $B$ vertically stacked, but separated by a dashed line. For increased clarity, we can draw boxes around functions. In Figure 5, we see a clear reexpression of Figure 4. Here, we have the unchanged $A$ variable untouched by $f$, which acts only on $B \times C$.

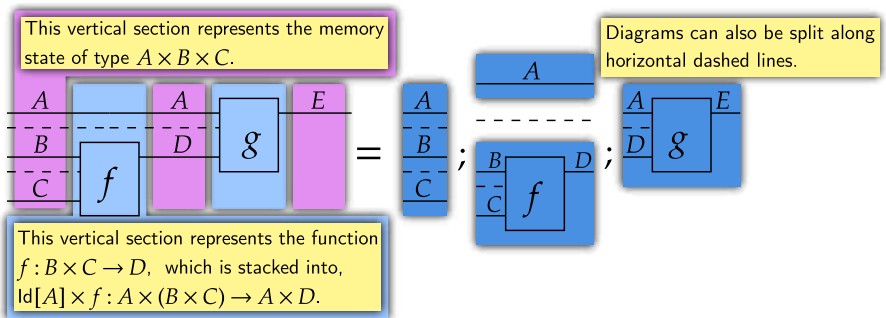

Figure 5: Tupled data types are diagrammed with wires separated by dashed lines. This clearly shows when functions act on only some variables.

Every vertical section of a diagram represents something. Either, it shows which data type is present in memory, or which function is applied at this step. Diagrams can always be decomposed into vertical sections, each of which must compose with adjacent sections to ensure algorithms are well-defined. Diagrams can also

be split along dashed lines. Diagrams are built from these vertically and horizontally composed sections, with wires acting like jigsaw indents.

### 2.3 Tensors

We will specialize our diagrams for deep learning models whose memory states are tuples of tensors. Tensors are numbers arranged along axes. So, a scalar $\mathbb{R}$ is a rank 0 tensor, a vector $\mathbb{R}^3$ is a rank 1 tensor, a table $\mathbb{R}^{4\times3}$ is a rank 2 tensor, and so on. If our diagram takes tensor data types, we get something like Figure 6.

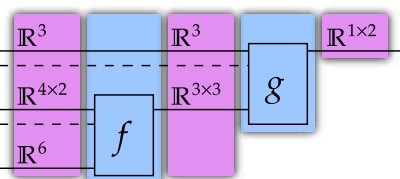

Figure 6: Similar to Figure 5, but with data types being tensors.

However, we benefit from diagramming the details of axes. Instead of diagramming a wire labeled $\mathbb{R}^{a\times b}$, we diagram a wire labeled $a$ and a wire labeled $b$, without a dashed line separating them. This lets us diagram Figure 6 into the clear form of Figure 7.

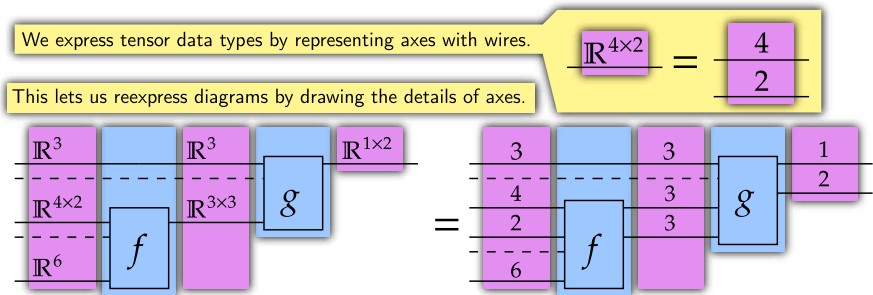

Figure 7: We can diagram types $\mathbb{R}^{a\times b}$ as two wires labeled $a$ and $b$, without a dashed line separating them. *(See cell 2, Jupyter notebook.)*

#### 2.3.1 Indexes

Values in tensors are accessed by indexes. A tensor $A \in \mathbb{R}^{4\times3}$, for example, has constituent values $A[i_4, j_3] \in \mathbb{R}$, where $i_4 \in \{0\ldots3\}$ and $j_3 \in \{0\ldots2\}$. Indexes can also be used to access subtensors, so we have expressions $A[i_4, :] \in \mathbb{R}^3$. This subtensor extraction is therefore an operation $\mathbb{R}^{4\times3} \to \mathbb{R}^3$. We diagram it by having indexes act on the relevant axis. Indexes are diagrammed with pointed pentagons, or kets $|\_\rangle$. This type of subtensor extraction is diagrammed according to Figure 8.

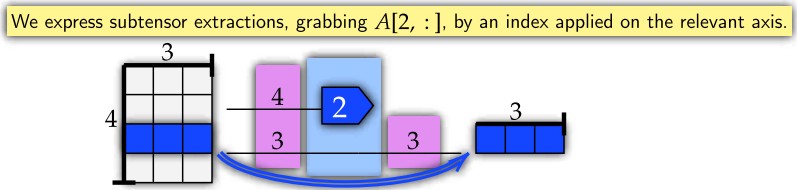

Figure 8: We diagram indexes with pointed pentagons labeled with the index being extracted. *(See cell 3, Jupyter notebook.)*

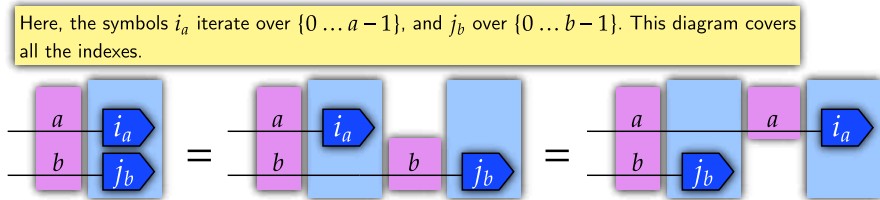

Figure 9: These subtensors are defined such that $A[i_4, :][j_3] = A[i_4, j_3]$. This expression is the same in the reverse order. *(See cell 4, Jupyter notebook.)*

### 2.3.2 Broadcasting

Broadcasting is critical to understanding deep learning models. It lifts an operation to act in parallel over additional axes. Here, we show an operation $G : \mathbb{R}^3 \to \mathbb{R}^2$ lifted to an operation $G' : \mathbb{R}^{4 \times 3} \to \mathbb{R}^{4 \times 2}$. We diagram this broadcasting by having the 4-length wire pass over $G$, adding a 4-length axis to its input and output shapes. Formally, we define $G'(x)[i_4, :] = G(x[i_4, :])$. This is shown in Figure 10.

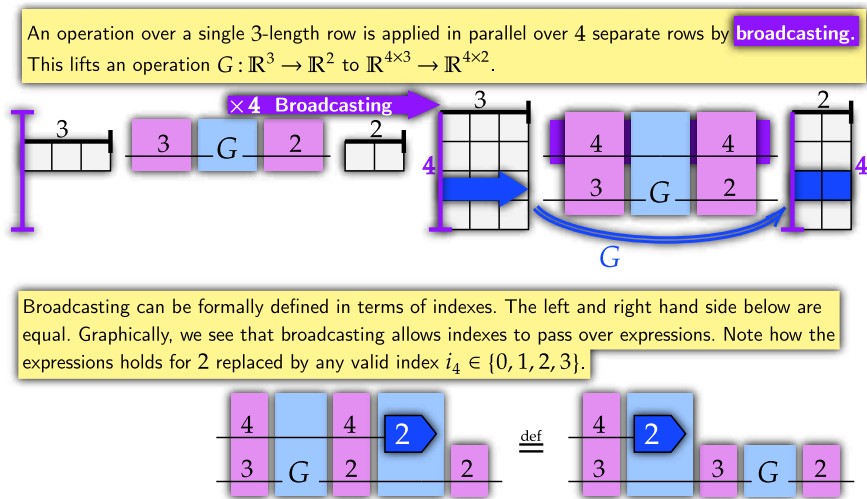

Figure 10: An operation is lifted over a 4-length axis by broadcasting. This applies $G$ over corresponding subtensors. Broadcasting can be formally defined by equating indexes before and after an operation. *(See cell 5, Jupyter notebook.)*

Inner broadcasting acts within tuple segments. A $\mathbb{R}^{4 \times 3} \times \mathbb{R}^4$ collection of data can be reduced to $\mathbb{R}^3 \times \mathbb{R}^4$ in 4 different ways. Therefore, there are 4 ways of applying an operation $H : \mathbb{R}^3 \times \mathbb{R}^4 \to \mathbb{R}^2$ to it. This gives a function lifted by "inner broadcasting", which has a shape $\mathbb{R}^{4 \times 3} \times \mathbb{R}^4 \to \mathbb{R}^{4 \times 2}$. We diagram this by drawing a wire from the source tuple segment over the function, as shown in Figure 15. This adds an axis of equal length to the target tuple segment and to the output, reflecting the shape of the lifted operation.

Broadcasting naturally represents element-wise operations. A function on values $f : \mathbb{R}^1 \to \mathbb{R}^1$, when broadcast, gives an operation $\mathbb{R}^{1 \times a} \to \mathbb{R}^{1 \times a}$. One length axes do not change the shape of data, and can be freely amended or removed from pre-existing shapes by arrows. This means we diagram element-wise functions by drawing incoming and outgoing arrows, which represent the amendment and removal of a 1-length axis. This is shown in Figure 12.

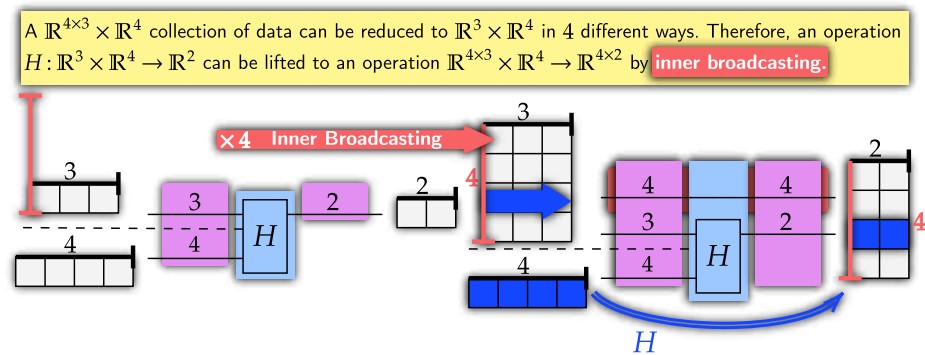

Figure 11: Lifting an operation within a tuple segment gives inner broadcasting. We diagram it by having a wire from the target tuple segment over the function, reflecting the shape of the lifted function. *(See cell 6, Jupyter notebook.)*

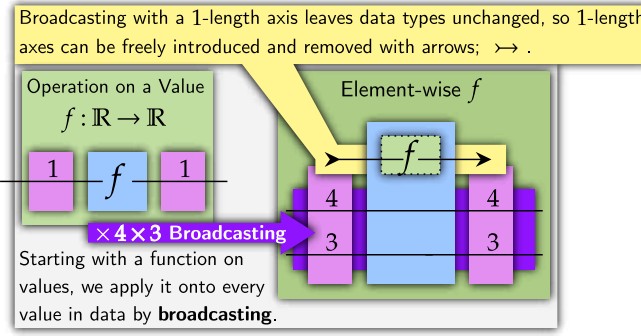

Figure 12: Element-wise operations can be naturally shown with broadcasting. *(See cell 7, Jupyter notebook.)*

## 2.4 Linearity

Linear functions are an important class of operations for deep learning. Linear functions can be highly parallelized, especially with GPUs. Previous works have shown how graphically modeling linear functions, and reimplementing algorithms can improve performance (Xu et al., 2023). Linear functions have immense regularity. Standard monoidal string diagrams rely on these properties to provide elegant graphical languages for various fields (Baez & Stay, 2010).

Linear functions are required to obey additivity and homogeneity, as shown in Figure 13. These operations are closed under composition, so applying linear maps onto each other gives another linear map. Importantly to us, they are natural with respect to broadcasting. This means for any two linear functions $f$ and $g$, the equality in Figure 14 holds. This means they can be simultaneously broadcast. This lets a series of linear functions be efficiently parallelized and flexibly rearranged.

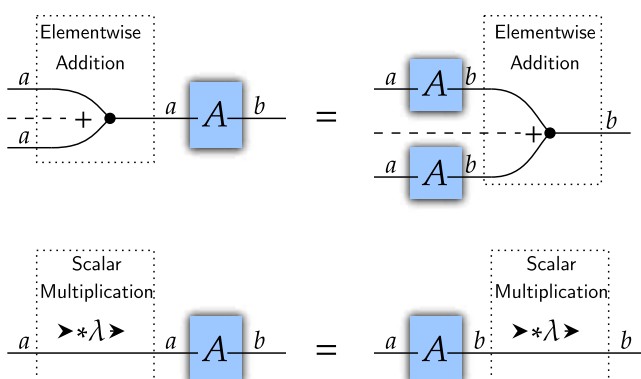

Figure 13: A subset of functions between $\mathbb{R}^a$ to $\mathbb{R}^b$ are linear, obeying additivity and homogeneity. This class of functions are closed under composition and has many important composition properties.

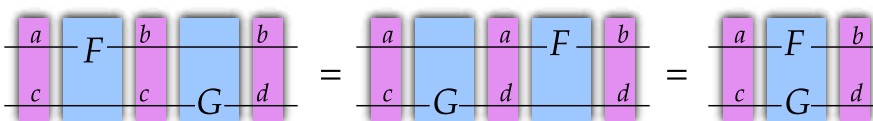

Figure 14: Linear functions are natural with respect to each other and broadcasting. This means the above equality holds, letting expressions be flexibly rearranged.

However, a pure monoidal string diagram has difficulty representing non-linear operations, a noted issue (Chiang et al., 2023). Neural circuit diagrams have Cartesian products and broadcasting, which are not generally analogous to how monoidal string diagrams combine linear functions. If we know functions are linear, we can use diagrams to efficiently reason about algorithms. By focusing on linear functions, we can take advantage of their parallelization properties.

### 2.4.1 Multilinearity

There is an important distinction between linear and multilinear operations. Inner products, for example, are multilinear. The inner product $u(\mathbf{x}, \mathbf{y}) = \mathbf{x} \cdot \mathbf{y} = \Sigma_i \, x[i] \cdot y[i]$ is linear with respect to each input. So, $u(\mathbf{x} + \mathbf{z}, \mathbf{y}) = u(\mathbf{x}, \mathbf{y}) + u(\mathbf{z}, \mathbf{y})$, and similarly for the second input. However, it is not linear with respect to element-wise addition over its entire input and output, as $u(\mathbf{x}_1 + \mathbf{x}_2, \mathbf{y}_1 + \mathbf{y}_2) \neq u(\mathbf{x}_1, \mathbf{y}_1) + u(\mathbf{x}_2, \mathbf{y}_2)$. Compare this to copying $\Delta$, which we can show is linear.

$$\Delta : \mathbb{R}^a \to \mathbb{R}^{a \times a} \text{ and } \mathbf{x}, \mathbf{y} \in \mathbb{R}^a, \ \lambda \in \mathbb{R}$$
$$\Delta(\mathbf{x}) := (\mathbf{x}, \mathbf{x})$$
$$\Delta(\mathbf{x} + \mathbf{y}) = (\mathbf{x} + \mathbf{y}, \mathbf{x} + \mathbf{y}) = (\mathbf{x}, \mathbf{x}) + (\mathbf{y} + \mathbf{y})$$
$$= \Delta(\mathbf{x}) + \Delta(\mathbf{y})$$
$$\Delta(\lambda \cdot \mathbf{x}) = (\lambda \cdot \mathbf{x}, \lambda \cdot \mathbf{x}) = \lambda \cdot (\mathbf{x}, \mathbf{x})$$
$$= \lambda \cdot \Delta(\mathbf{x})$$

To simultaneously broadcast multilinear functions, we note that every multilinear operation equals an outer product followed by a linear function. The outer product is the ur-multilinear operation, taking a tuple input and returning a tensor, which takes the product over one element from each tuple segment. It is given by $\otimes : \mathbb{R}^a \times \mathbb{R}^b \to \mathbb{R}^{a \times b}$. All tuple-multilinear functions $M : \mathbb{R}^a \times \mathbb{R}^b \to \mathbb{R}^c$ have an associated tensor-linear form $M_\lambda : \mathbb{R}^{a \times b} \to \mathbb{R}^c$ such that $\otimes; M_\lambda = M$. We diagram the outer product by simply having a tuple line ending, which will often occur before a host of linear operations are simultaneously applied.

### 2.4.2 Implementing Linearity and Common Operations

Key linear and multilinear operations can be implemented by the einops package (Rogozhnikov, 2021), leading to elegant implementations of algorithms. Some key linear operations are **inner products**, which sum over an axis, **transposing**, which swaps axes, **views**, which rearranges axes, and **diagonalization**, which makes axes take the same index.

With neural circuit diagrams, we can clearly show these operations. We show inner products with cups, transposing by crossing wires, views by solid lines consuming and producing their respective shapes, and diagonalization by wires merging. As these operations are linear, they can be simultaneously applied. The interaction of wires shows how incoming axes coordinate to produce outgoing axes. The einops package symbolically implements these operations by having incoming and outgoing axes correspond to symbols.

An example that combines many of these operations is a section of multi-head attention shown in 16. It employs an outer product, a transpose, a diagonalization, an inner product, and an element-wise operation. The input to this algorithm is a tuple of tensors. Axes with an overline are a width, representing the amount of rather than detail per thing. Though a complex expression, we can break this figure up as in Figure 7 and implement the interaction of wires using einops, shown in Figure 17.

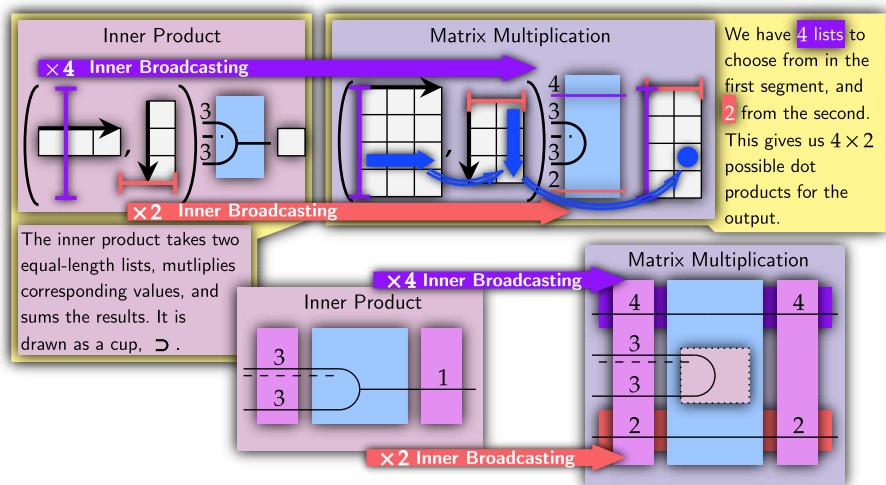

Figure 15: An in-depth example of matrix multiplication, a key multilinear inner broadcast operation. Inner products are defined on vectors $\mathbb{R}^n \times \mathbb{R}^n \to \mathbb{R}^1$. Then, we inner broadcast them to act over matrices. The new $\mathbb{R}^{p \times n} \times \mathbb{R}^{n \times q} \to \mathbb{R}^{p \times q}$ operation is matrix multiplication. Therefore, we see that matrix multiplication is an instance of an inner broadcast operation.

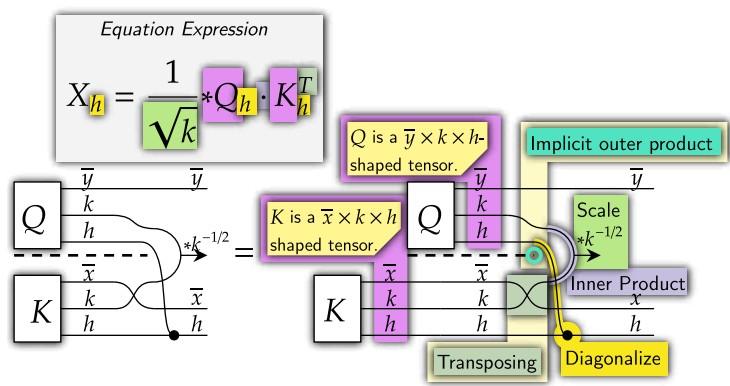

Figure 16: We can diagram a portion of multi-head attention, a sophisticated algorithm, with clarity using neural circuit diagrams.

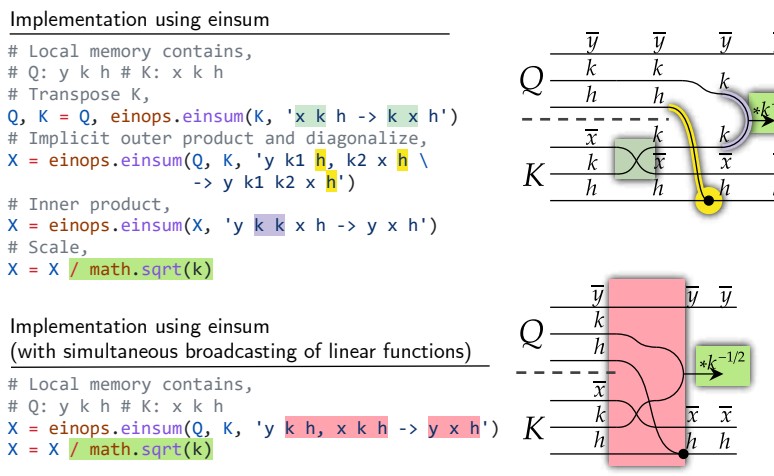

Figure 17: This section of multi-head attention can be implemented using the einsum operation. Note the close relationship between diagrams and implementation and how diagrams reflect the memory states and operations of algorithms. *(See cell 8, Jupyter notebook.)*

### 2.4.3   Linear Algebra

All linear functions $f : \mathbb{R}^a \to \mathbb{R}^b$ have an associated $\mathbb{R}^{a \times b}$ tensor that uniquely identifies them. This hints at the ability to transpose this associated tensor to get a new linear function, $f^T : \mathbb{R}^b \to \mathbb{R}^a$. To extract these associated transposes, we use the unit. The **unit** for a shape $a$, given by $\eta : \mathbb{R}^1 \to \mathbb{R}^{a \times a}$, is a linear map which returns $r$ times the $\mathbb{R}^{a \times a}$ identity matrix, for $r \in \mathbb{R}$.

Note that the associated transpose, which sends a linear function $f : \mathbb{R}^n \to \mathbb{R}^m$ to $f^T : \mathbb{R}^m \to \mathbb{R}^n$ by transposing the associated $\mathbb{R}^{n \times m}$ tensor, is different to a transpose operation which sends $\mathbb{R}^{n \times m}$ to $\mathbb{R}^{m \times n}$. Associated transposes are used for mathematical rearrangement and are not usually directly implemented in code, though I provide code examples in cell 9 of the Jupyter notebook.

The unit and the inner product can be arranged to give the identity map $\mathbb{R}^a \to \mathbb{R}^a$, as in Figure 18. This identity map can be freely introduced, split into a unit and the identity matrix, and then used to rearrange operations. For example, this allows us to convert the linear map $F : \mathbb{R}^a \to \mathbb{R}^{b \times c}$ into $F^T : \mathbb{R}^{b \times a} \to \mathbb{R}^c$. These associated tensors and transposes can be used to better understand convolution (Section 3.3) and backpropagation (Section 3.6).

Figure 18: Linear operations have a flexible algebra. Simultaneous operations may increase efficiency (Xu et al., 2023). As the height of diagrams is related to the amount of data stored in independent segments, it gives a rough idea of memory usage. This is further explored in Section 3.6.   *(See cell 9, Jupyter notebook.)*

These rearrangements can transpose specific axes. A linear operation $\mathbb{R}^{a \times b} \to \mathbb{R}^c$ has an associated $\mathbb{R}^{a \times b \times c}$ tensor. This tensor can be associated with various linear operations, such as $\mathbb{R}^{b \times a} \to \mathbb{R}^c$. These different forms are often of interest to us, as they can efficiently implement the reverse of operations (see Figure 25, 30). To extract these rearrangements, we can selectively apply units and the inner product to reorient the direction of wires for linear operations.

# 3 Results: Key Applied Cases

## 3.1 Basic Multi-Layer Perceptron

Diagramming a basic multi-layer perceptron will help consolidate knowledge of neural circuit diagrams and show their value as a teaching and implementation tool, as shown in Figure 19. We use pictograms to represent components analogous to traditional circuit diagrams and to create more memorable diagrams (Borkin et al., 2016).

```python
import torch.nn as nn
# Basic Image Recogniser
# This is a close copy of an introductory PyTorch tutorial:
# https://pytorch.org/tutorials/beginner/basics/buildmodel_tutorial.html
class BasicImageRecogniser(nn.Module):
    def __init__(self):
        super().__init__()
        self.flatten = nn.Flatten()
        self.linear_relu_stack = nn.Sequential(
            nn.Linear(28*28, 512),
            nn.ReLU(),
            nn.Linear(512, 512),
            nn.ReLU(),
            nn.Linear(512, 10),
        )
    def forward(self, x):
        x = self.flatten(x)
        x = self.linear_relu_stack(x)
        y_pred = nn.Softmax(x)
        return y_pred
```

Figure 19: PyTorch code and a neural circuit diagram for a basic MNIST (digit recognition) neural network taken from an introductory PyTorch tutorial. Note the close correspondence between neural circuit diagrams and PyTorch code. *(See cell 10, Jupyter notebook.)*

Fully connected layers are shown as boldface **L**, with boldface indicating a component with internal learned weights. Their input and output sizes are inferred from the diagrams. If a fully connected layer is biased, we add a "**+**" in the bottom right. Traditional presentations easily miss this detail. For example, many implementations of the transformer, including those from PyTorch and Harvard NLP, have a bias in the query, key, and value fully-connected layers despite *Attention is All You Need* (Vaswani et al., 2017) not indicating the presence of bias.

Activation functions are just element-wise operations. Though traditionally ReLU (Krizhevsky et al., 2012), other choices may yield superior performance (Lee, 2023). With neural circuit diagrams, the activation function employed can be checked at a glance. SoftMax is a common operation that converts scores into probabilities, and we represent it with a left-facing triangle ($\triangleleft$), indicating values being "spread" to sum to 1.

As mentioned in Section 1.2, how operations such as SoftMax are broadcast can be ambiguous in traditional presentations. This is especially worrisome as SoftMax can be applied to shapes of arbitrary size. On the other hand, the neural circuit diagram method of displaying broadcasting makes it clear how SoftMax is applied.

### 3.2 Neural Circuit Diagrams for the Transformer Architecture

In Section 1.2, we covered shortfalls in *Attention is All You Need*. We now have the tools to address these shortcomings using neural circuit diagrams. Figure 20 shows scaled-dot product attention. Unlike the approach from *Attention is All You Need*, the size of variables and the axes over which matrix multiplication and broadcasting occur is clearly shown. Figure 21 shows multi-head attention. The origin of queries, keys, and values are clear, and concatenating the separate attention heads using einsum naturally follows. Finally, we show the full transformer model in Figure 22 using neural circuit diagrams. Introducing such a large architecture requires an unavoidable level of description, and we take some artistic license and notate all the additional details.

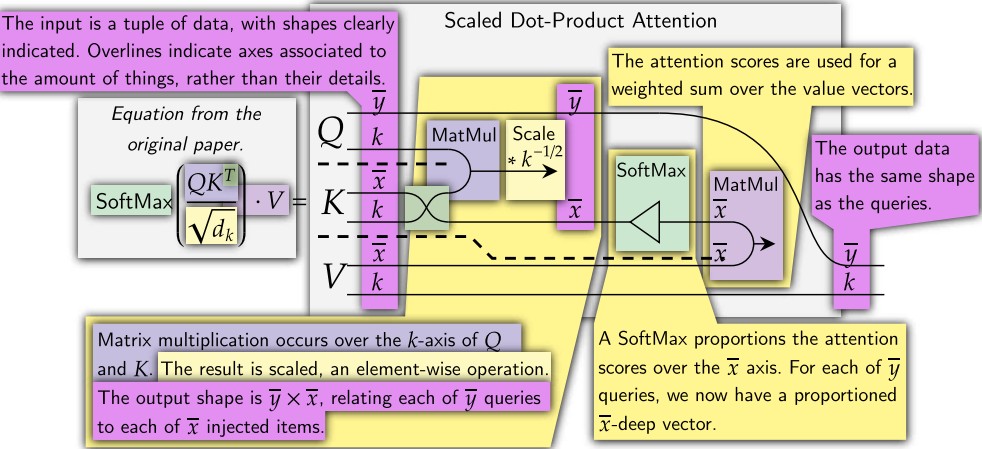

Figure 20: The original equation for attention against a neural circuit diagram. The descriptions are unnecessary but clarify what is happening. Corresponds to Equation 1 and Figure 1.1. *(See cell 11, Jupyter notebook.)*

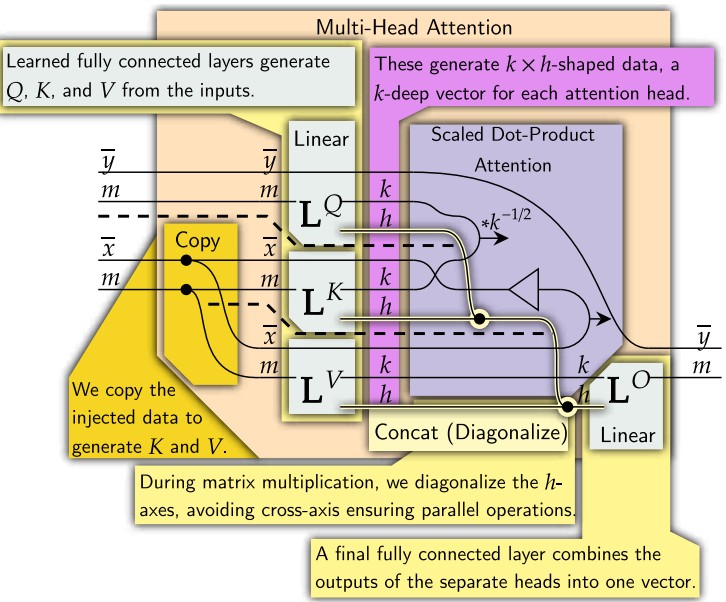

Figure 21: Neural circuit diagram for **multi-head attention**. Implementing matrix multiplication is clear with the cross-platform the einops package (Rogozhnikov, 2021). Corresponds to Equation 2 and 3 and Figure 1.2. *(See cell 12, Jupyter notebook.)*

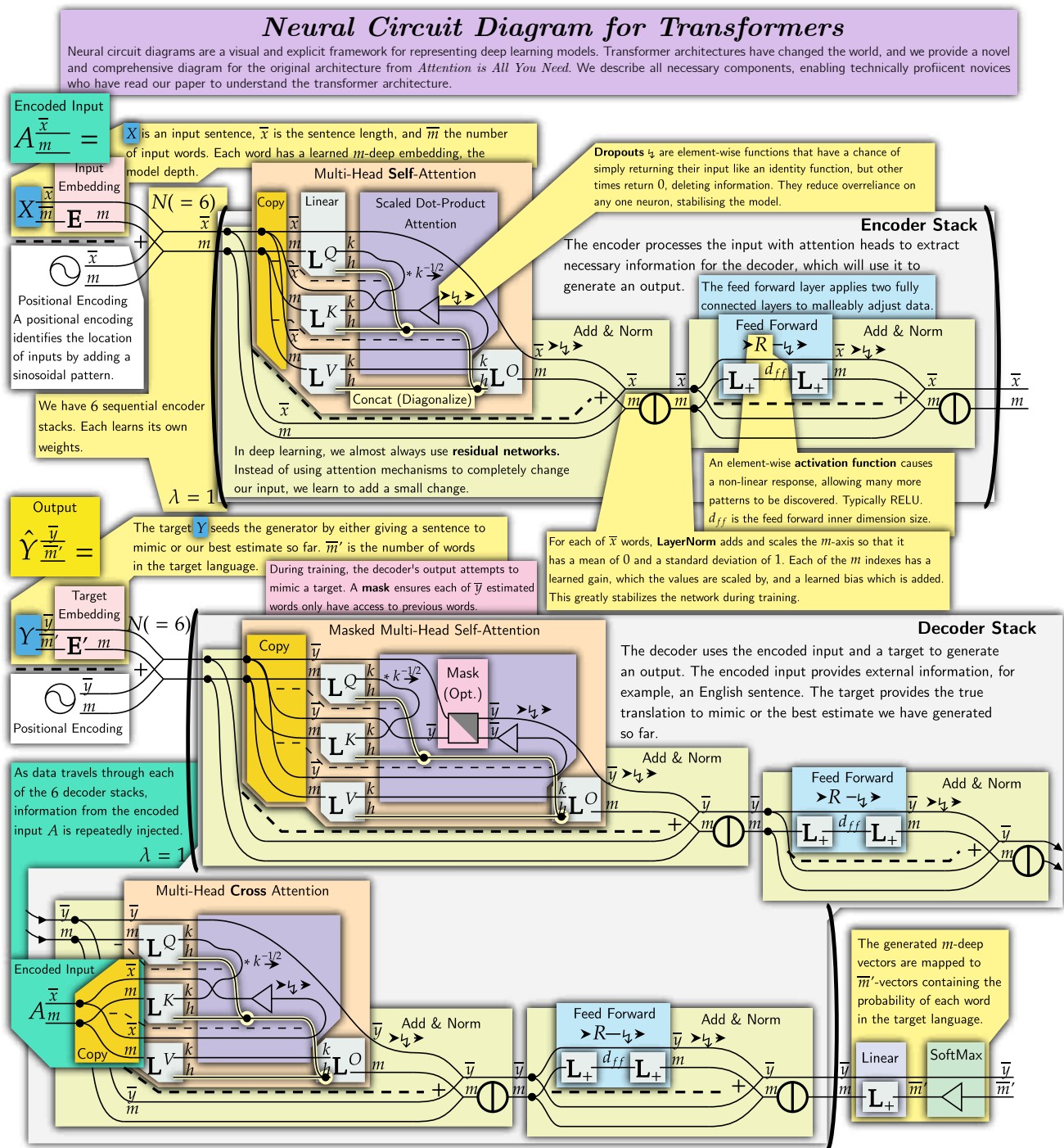

Figure 22: The fully diagrammed architecture from *Attention is All You Need* (Vaswani et al., 2017).

### 3.3 Convolution

Convolutions are critical to understanding computer vision architectures. Different architectures extend and use convolution in various ways, so implementing and understanding these architectures requires convolution and its variations to be accurately expressed. However, these extensions are often hard to explain. For example, PyTorch concedes that dilation is "harder to describe". Transposed convolution is similarly challenging to communicate (Zeiler et al., 2010). A standardized means of notating convolution and its variations would aid in communicating the ideas already developed by the machine learning community and encourage more innovation of sophisticated architectures such as vision transformers (Dosovitskiy et al., 2021; Khan et al., 2022).

In deep learning, convolutions alter a tensor by taking weighted sums over nearby values. With standard bracket notation to access values, a convolution over vector $v$ of length $\overline{x}$ by a kernel $w$ of length $k$ is given by, *(Note: we subscript indexes by the axis over which they act.)*

$$\text{Conv}(v, w)[i_{\overline{y}}] = \sum_{j_k} v[i_{\overline{y}} + j_k] \cdot w[j_k]$$

The maximum $i_{\overline{y}}$ value is such that it does not exceed the maximum index for $v[i_{\overline{y}} + j_k]$. Starting indexing at 0, we get $\overline{x} - 1 = i_{\max} + j_{\max} = \overline{y} + k - 2$, so the length of the output is therefore $\overline{y} = \overline{x} - k + 1$. Note how convolution is a multilinear operation; it is linear concerning each vector input $v$ and $w$. Therefore, it has a tensor-linear form with an associated tensor, the convolution tensor, that uniquely identifies it.

$$\text{Conv}(v, w)[i_{\overline{y}}] = \sum_{j_k} \sum_{\ell_x} (\star)[i_{\overline{y}}, j_k, \ell_{\overline{x}}] \cdot v[\ell_{\overline{x}}] \cdot w[j_k]$$

$$(\star)[i_{\overline{y}}, j_k, \ell_{\overline{x}}] = \begin{cases} 1 & \text{, if } \ell_{\overline{x}} = i_{\overline{y}} + j_k. \\ 0 & \text{, else.} \end{cases}$$

We diagram convolution with the below diagram, Figure 23. We then transpose the linear operation into a more standard form, letting the input be to the left, and the kernel be to the right.

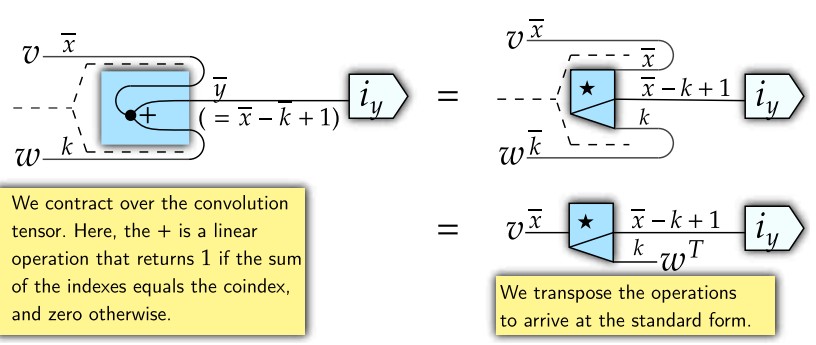

We contract over the convolution tensor. Here, the + is a linear operation that returns 1 if the sum of the indexes equals the coindex, and zero otherwise.

We transpose the operations to arrive at the standard form.

Figure 23: Convolution is a multilinear operation, with an associated tensor. This tensor is transposed into a standard form.

We typically work with higher dimensional convolutions, in which case the indexes act like tuples of indexes. We diagram axes that act in this tandem manner by placing them especially close to each other and labeling their length by one bolded symbol akin to a vector. In 2 dimensions the convolution tensor becomes;

$$(\star\ 2D)[i_{\overline{y0}}, i_{\overline{y1}}, j_{k0}, j_{k1}, \ell_{\overline{x0}}, \ell_{\overline{x1}}] = \begin{cases} 1 & \text{, if } (\ell_{\overline{x0}}, \ell_{\overline{x1}}) = (i_{\overline{y0}}, i_{\overline{y1}}) + (j_{k0}, j_{k1}). \\ 0 & \text{, else.} \end{cases}$$

Figure 24 shows what convolution does. It takes an input, uses a linear operation to separate it into overlapping blocks, and then broadcasts an operation over each block. Using neural circuit diagrams, we

now easily show the extensions of convolution. A standard convolution operation tensors the input with a channel depth axis, and feeds each block and the channel axis through a learned linear map.

Additionally, we can take an average, maximum, or some other operation rather than a linear map on each block. This lets us naturally display average or max pooling, among other operations. Displaying convolutions like this has further benefits for understanding. For example, $1 \times 1$ convolution tensors give a linear operation $\mathbb{R}^{\overline{\mathbf{x}}} \to \mathbb{R}^{\overline{\mathbf{x}} \times 1}$, which we recognize to be the identity. Therefore, $1 \times 1$ kernels are the same as broadcasting over the input.

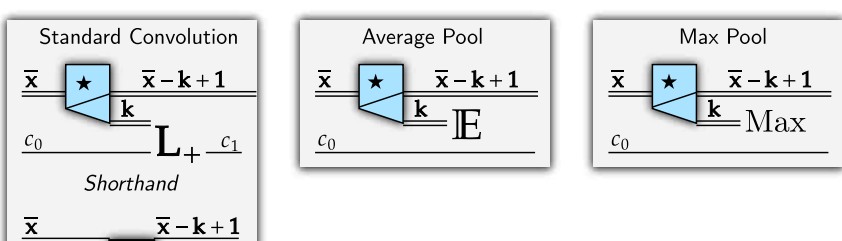

Figure 24: Convolution and related operations, clearly shown using neural circuit diagrams.

*Stride* and *dilation* scale the contribution of $i_y$ or $j_k$ in the convolution tensor, increasing the speed at which the convolution scans over its inputs. This changes the convolution tensor into the form of Equation 4. We diagram these changes by adding the $s$ or $d$ multiplier where the axis meets the tensor as in Figure 25. These multipliers also change the size of the output, allowing for downscaling operations.

$$(\star \ s, d)[i_{\overline{y}}, j_k, \ell_{\overline{x}}] = \begin{cases} 1 & \text{, if } \ell_{\overline{x}} = s \ * \ i_{\overline{y}} + d \ * \ j_k. \\ 0 & \text{, else.} \end{cases} \tag{4}$$

$$\overline{y} = \left\lfloor \frac{\overline{x} - d \ * \ (k-1) - 1}{s} + 1 \right\rfloor \tag{5}$$

We often want to make slight adjustments to the output size. This is done by **padding** the input with zeros around its borders. We can explicitly show the padding operation, but we make it implicit when the output dimension does not match the expectation given the input dimension, kernel dimension, stride, and dilation used.

Stride can make the output axis have a far lower dimension than the input axis. This is perfect for down-scaling. We implement upscaling by transposing strided convolution, resulting in an operation with many more output blocks than actual inputs. We broadcast over these blocks to get our high-dimensional output.

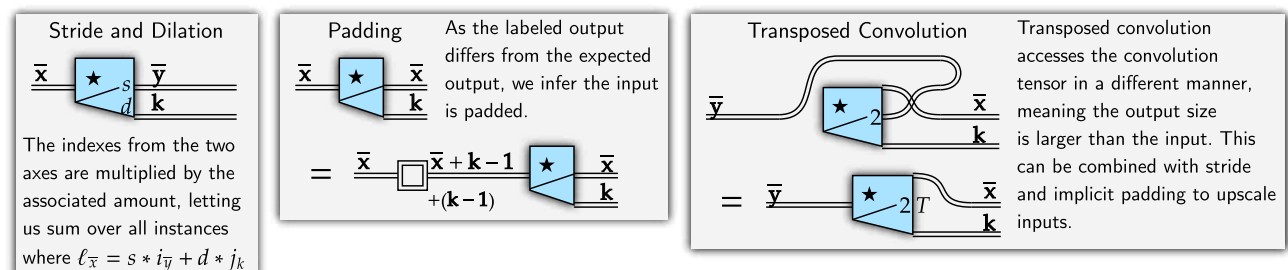

Figure 25: Stride, dilation, padding, and transposed convolution shown with neural circuit diagrams.

Transposed convolution is challenging to intuit in the typical approach to convolutions, which focuses on visualizing the scanning action rather than the decomposition of an image's data structure into overlapping blocks. The blocks generated by transposed convolution can be broadcast with linear maps, maximum, average, or other operations, all easily shown using neural circuit diagrams.

### 3.4 Computer Vision

In computer vision, the design of deep learning architectures is critical. Computer vision tasks often have enormous inputs that are only tractable with a high degree of parallelization (Krizhevsky et al., 2012). Architectures can relate information at different scales (Luo et al., 2017), making architecture design task-dependant. Sophisticated architectures such as vision transformers combine the complexity of convolution and transformer architectures (Khan et al., 2022; Dehghani et al., 2023).

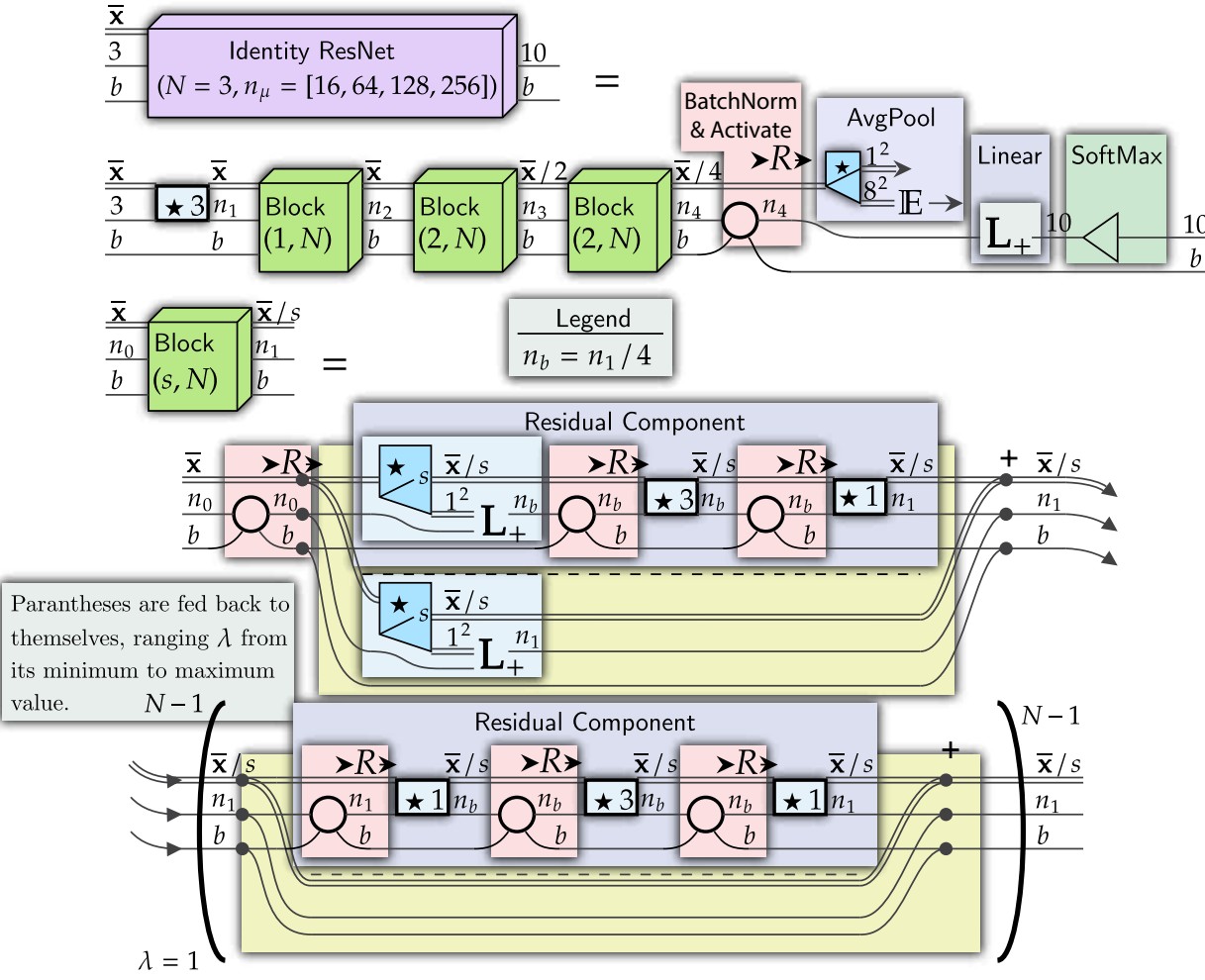

Figure 26: Residual networks with identity mappings and full pre-activation (IdResNet) (He et al., 2016) offered improvements over the original ResNet architecture. These improvements, however, are often missing from implementations. By making the design of the improved model clear, neural circuit diagrams can motivate common packages to be updated. (*See cell 13-15, Jupyter notebook.*)

These cases show why clear architecture design is promising for enhancing computer vision research. Neural circuit diagrams, therefore, are in a unique position to accelerate computer vision research, motivating parallelization, task-appropriate architecture design, and further innovation of sophisticated architectures.

As examples of neural circuit diagrams applied to computer vision architectures, I have diagrammed the identity residual network architecture (He et al., 2016) in Figure 26, which shows many innovations of ResNets not included in common implementations, as well as the UNet architecture (Ronneberger et al., 2015) in Figure 27, which shows how saving and loading variables may be displayed.

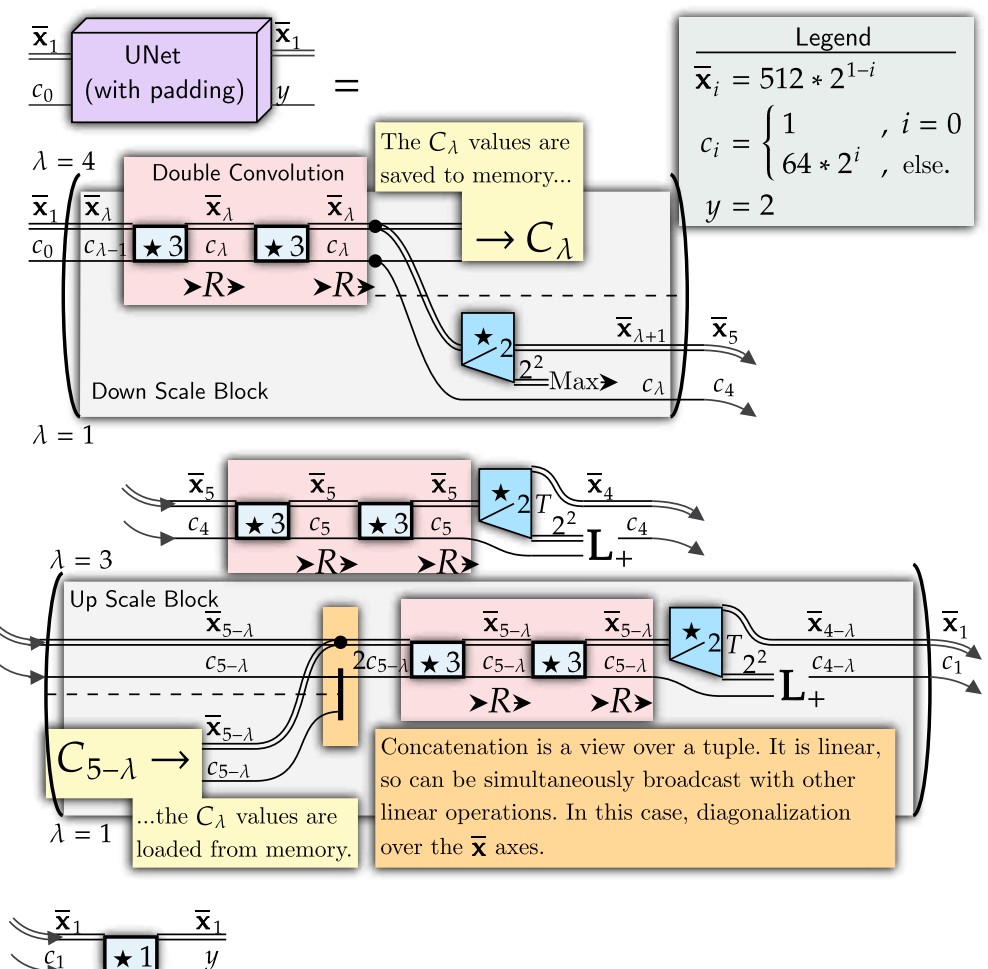

Figure 27: The UNet architecture (Ronneberger et al., 2015) forms the basis of probabilistic diffusion models, state-of-the-art image generation tools (Rombach et al., 2022). UNets rearrange data in intricate ways, which we can show with neural circuit diagrams. Note that in this diagram we have modified the UNet architecture to pad the input of convolution layers. To get the original UNet architecture, the $\overline{\mathbf{x}}_\lambda$ values can be further distinguished as $\overline{\mathbf{x}}_{\lambda,j}$, the sizes of which can be added to the legend. *(See cell 16, Jupyter notebook.)*

Architectures often comprise sub-components, which we show as blocks that accept configurations. This is analogous to classes or functions that may appear in code. The code associated with this work implements these algorithms guided by the blocks from the diagrams.

### 3.5 Vision Transformer

Neural circuit diagrams reveal the degrees of freedom of architectures, motivating experimentation and innovation. A case study that reveals this is the vision transformer, which brings together many of the cases we have already covered. Its explanations (Khan et al., 2022, See Figure 2) suffer from the same issues as explanations of the original transformer (see Section 1.2), made worse by even more axes being present.

With neural circuit diagrams, visual attention mechanisms are as simple as replacing the $\overline{y}$ and $\overline{x}$ axes in Figure 21 with tandem $\overline{\mathbf{y}}$ and $\overline{\mathbf{x}}$ axes and setting $h = 1$. As $1 \times 1$ convolutions are simply the identity map, $\mathrm{Conv}(v, [1]) = v$, broadcasting a linear map $\mathbb{R}^c \to \mathbb{R}^k$ for each of $\overline{\mathbf{y}}$ pixels is a $1 \times 1$-convolution. This leaves us with Figure 28 for a visual attention mechanism.

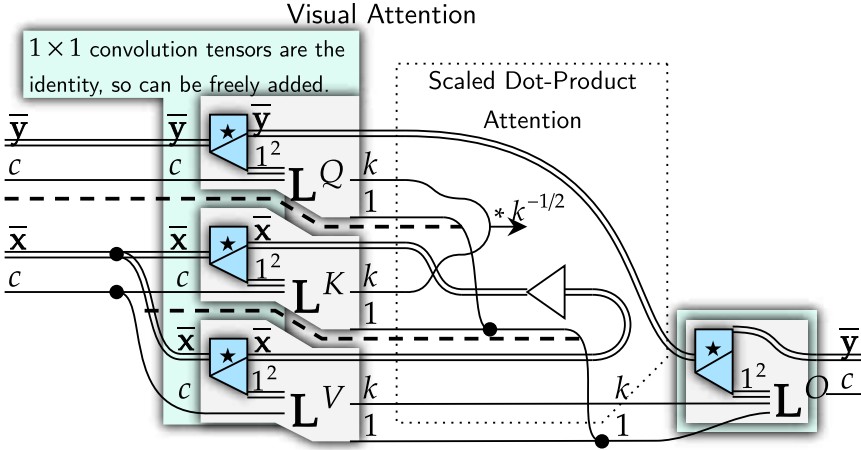

Figure 28: Using neural circuit diagrams, visual attention (Dosovitskiy et al., 2021) is shown to be a simple modification of multi-head attention (*See Figure 21, Figure 16, cell 17, Jupyter notebook.*)

This highly suggestive diagram calls us to experiment with the convolutions' stride, dilation, and kernel sizes, potentially streamlining models. The diagram clarifies how to implement multi-head visual attention with $h \neq 1$, especially using einsum similar to Figure 16. Additionally, $\overline{\mathbf{y}}$ does not need to match $\overline{\mathbf{x}}$. We could have $\overline{\mathbf{y}}$ be image data, and $\overline{x}$ be textual data without convolutions.

This case study shows how neural circuit diagrams reveal the degrees of freedom of architectures and, therefore, motivate innovation while being precise in how algorithms should be implemented.

### 3.6 Differentiation: A Clear Improvement over Prior Methods

I leave the most mathematically dense part of this work for last. Neural circuit diagrams intend to be used for the communication, implementation, tinkering, and analysis of architectures. These aims appeal to distinct audiences, and each should conceptualize neural circuit diagrams differently. The theoretical study of deep learning models requires understanding how individual components are composed into models and how properties scale during composition. Neural circuit diagrams are highly composed systems (see Figure 7) and thus provide a framework for studying composition. They have an underlying category, which is not the focus of this work.

Differentiation is an example of a property that is agreeable under composition. Differentiation is key to understanding information flows through architectures (He et al., 2016). The chain rule relates the derivative of composed functions to the composition of their derivatives and, therefore, provides a case study of how studying composition allows models to be understood. This analysis, however, is hampered by the fact that symbolically expressing the chain rule has quadratic length complexity relative to the number of composed

functions.

$$h'(x) = h'(x)$$
$$(g \circ h)'(x) = (g' \circ h)(x) \cdot h'(x)$$
$$(f \circ g \circ h)'(x) = (f' \circ g \circ h)(x) \cdot (g' \circ h)(x) \cdot h'(x)$$

This issue of symbolic methods proliferating symbols to keep track of relationships between objects was noted in the introduction. To understand how differentiation is composed and encourage more innovations like that of identity ResNets, which used differentiation to understand data flows (He et al., 2016), we need a graphical differentiation method.

Some graphical methods have been developed and applied to understanding differentiation in the context of deep learning, drawing on monoidal string diagrams from category theory (Shiebler et al., 2021; Cockett et al., 2019). As linearity cannot be completely ensured, these graphical methods are Cartesian, not expressing the details of axes. Other graphical approaches to neural networks could not incorporate differentiation, showing the significance of neural circuit diagrams being able to incorporate differentiation (Xu & Maruyama, 2022).

Differentiation, however, has key linear properties. Transposing differentiation is very important. These prior graphical methods require redefining differentiation for each transpose, making the relationships between these forms unclear. By detailing tensors and Cartesian products, our graphical presentation can show these linear relationships clearly. While drawing on their many theoretical contributions (Shiebler et al., 2021; Cockett et al., 2019), this work provides a significant advantage over these previous works.

In addition to theoretical understanding, clearly expressing differentiation is key to efficient implementations. Mathematically equivalent algorithms may have different time or memory complexities. The rules of linear algebra we have developed (see Figure 18) allow mathematically equivalent algorithms to be rearranged into more time or memory-efficient forms. To show the potential of neural circuit diagrams, we focus on backpropagation and analyze its time and memory complexity with neural circuit diagrams.

### 3.6.1 Modeling Differentiation

To model differentiation, consider a once differentiable function $F : \mathbb{R}^a \rightarrow \mathbb{R}^b$. It has a Jacobian which assigns to every point in $\mathbb{R}^a$ a $\mathbb{R}^{a \times b}$ tensor that describes its derivative, $\mathsf{J}F : \mathbb{R}^a \rightarrow \mathbb{R}^{b \times a}$. Functions answer questions, and $\mathsf{J}F$ answers how much a function responds to an infinitesimal change. The questions we ask $\mathsf{J}F$ are *where* is the change happening ($\mathbb{R}^a$ input), *how* much is it changing by ($\mathbb{R}^b$ output axis), and *which* direction are we moving in ($\mathbb{R}^a$ output axis). Inner products over the output axes "ask" these questions. The chain rule can be defined with respect to the Jacobian and is diagrammed in Figure 29.

$$\left.\frac{\partial}{\partial x^a}(GF)^c\right|_{\mathbf{x}} = \sum_b \left( \left.\frac{\partial}{\partial x^b}G^c\right|_{F(\mathbf{x})} \cdot \left.\frac{\partial}{\partial x^a}F^b\right|_{\mathbf{x}} \right)$$

Figure 29: The chain rule expressed symbolically with index notation, and with neural circuit diagrams.

This expression is convoluted, and will struggle to scale. Instead, we transpose $\mathsf{J}F$ into the forward derivative as per Cockett et al. (2019)'s definition 4. This form is more agreeable for the chain rule, and is the first transpose we employ.

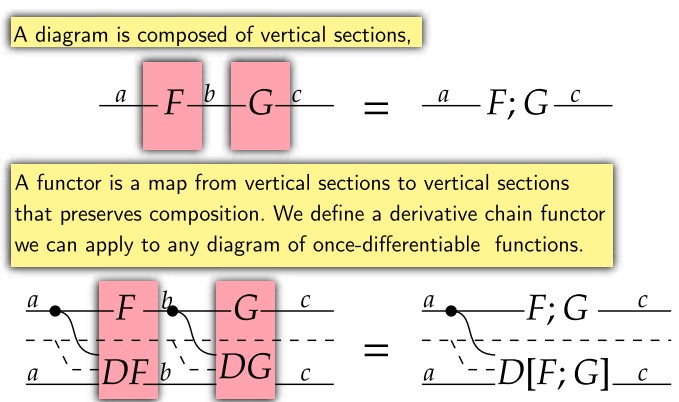

Figure 30: Definition of the forward derivative, and how functions compose under it.

This naturally scales with depth. Furthermore, we can define a $(\_, D\_)$ functor, a composition preserving map, from once differentiable functions $F : \mathbb{R}^a \to \mathbb{R}^b$ to $(F, DF) : \mathbb{R}^a \times \mathbb{R}^a \to \mathbb{R}^b \times \mathbb{R}^b$ (Fong et al., 2019; Cruttwell et al., 2021; Cockett et al., 2019). Per the chain rule, $(\_, D\_)[F; G] = (\_, D\_)F; (\_, D\_)G$. This is shown in Figure 31.

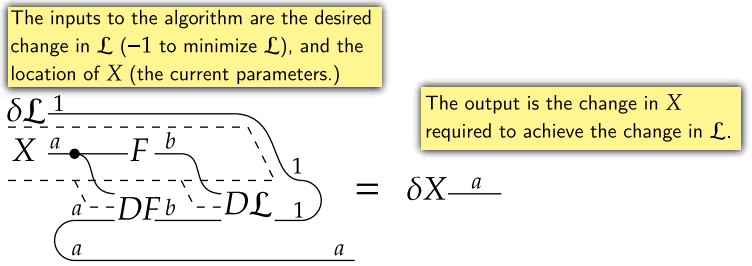

Figure 31: We have a composition preserving map, the $(\_, D\_)$ functor, that maps vertical sections to vertical sections, implementing the chain rule.

This composes elegantly. However, when optimizing an algorithm, we are not interested in how much a known infinitesimal change will alter an output. Rather, given some target change in output, we are interested in which direction will best achieve it. We can do this by calculating the change in the output for each element of $a$ in parallel, effectively running the algorithm multiple times. This is done by applying the unit and broadcasting. Furthermore, we sum the infinitesimal change over some target $\mathbb{R}^c$ value. The inner product does this. For an algorithm $F; \mathcal{L}$, where $\mathcal{L}$ is a loss function to $\mathbb{R}^1$, we can do optimization according to Figure 32.

Figure 32: We turn a small chain rule expression into an optimization function by applying the inner product over the target direction and derivative output. An inner product over an axis of length 1 is just multiplication. Using the unit, we run this algorithm for every input degree of freedom, broadcasting over the $a$ axis.

However, the forward derivative has large time complexity. A linear function gives matrix multiplication. Therefore, a linear map $f : \mathbb{R}^a \to \mathbb{R}^b$ applied onto $\mathbb{R}^a$ will require $a \times b$ operations. In general, broadcasting multiplies time and memory complexity. The memory usage of an algorithm is related to the number of elements it stores at any step in the algorithm. We use these tricks to analyze the order of the time and space complexity for the above process.

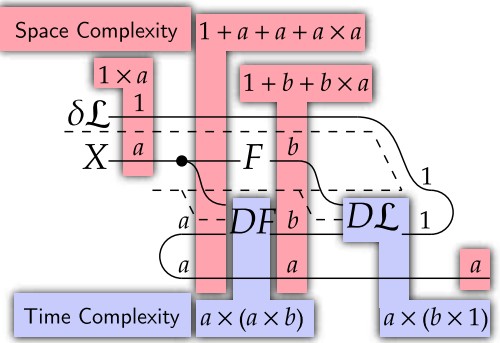

Figure 33: An analysis of the space and time complexity of the naive optimization algorithm.

We observe that this has a high time complexity, quadratic with respect to the size of $X$. In practice, we avoid the forward derivative, also called the Jacobian-vector product or JVP, in favor of the reverse derivative, or VJP, which more directly implements the above process. We define it in relation to the Jacobian and forward derivative in Figure 34. In Figure 36, we use our rules of linear algebra to re-express the optimization algorithm in terms of the forward derivative and show the far lower memory and time complexity required.

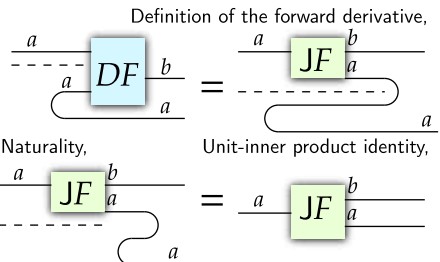

Figure 34: The definition of the forward and reverse derivative with respect to the Jacobian. This aligns with the Jacobian-vector product and the vector-Jacobian product, respectively.

Figure 35: A full expression of how the unit and the forward derivative give the Jacobian. This demonstrates how linear algebra principles can illustrate the relationships between different forms of the derivative.

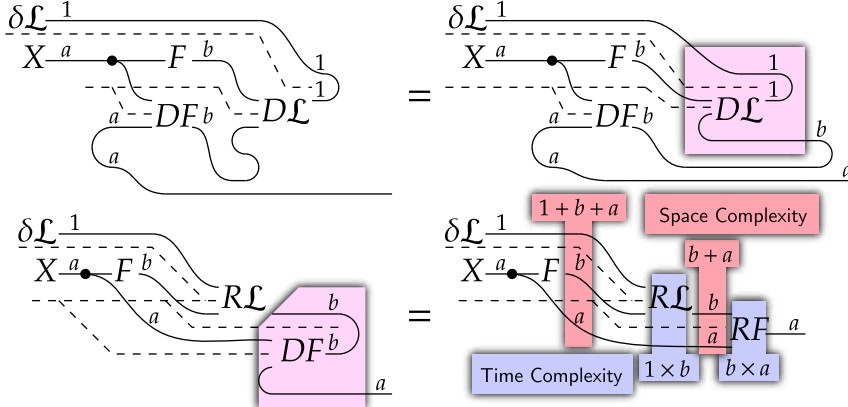

Figure 36: Using the previously developed linear algebra rules (see Figure 18, we rearrange our optimization algorithm to use the reverse instead of the forward derivative. The diagrams then reveal the computational advantages of the new algorithm, called backpropagation.

## 4   Conclusion

Neural circuit diagrams are a method of addressing the lingering problem of unclear communication in the deep learning community. My introduction showed why this is a concern and argued why a system of axis wires and dashed lines is needed, solving the challenge of reconciling the detail of tensor axes and the

flexibility of tuples. This work covered a host of architectures to breed familiarity, encourage adoption, and evidence the utility of neural circuit diagrams for understanding and implementing models.

Neural circuit diagrams are appealing to diverse users, from students first learning neural networks to specialized theoretical researchers investigating their mathematical foundations. This leads to immense future potential. Future work can see neural circuit diagrams explained in a concise and accessible manner for a lay audience. More diagrams can be modelled and formal standards developed . Finally, their mathematical foundation can be more fully expressed. The underlying category theory structure can be fully investigated (Abbott, 2023, Chapter 3), allowing models to incorporate probabilistic functions (Perrone, 2022; Fritz et al., 2023), additional data types, or even quantum circuits.

### Acknowledgements

Mathcha was used to write equations and draw diagrams. The Harvard NLP annotated transformer was invaluable for drawing Figure 22. I thank the anonymous TMLR reviewers for providing useful feedback on the paper throughout its many rewrites, and my supervisor Dr Yoshihiro Maruyama for his support, and pointing me towards applied category theory for machine learning. This work was supported by JST (JPMJMS2033-02; JPMJFR206P).

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

## A    Jupyter Notebook (see: vtabbott/Neural-Circuit-Diagrams)

```
import torch
import typing
import functorch
import itertools
```

## 2.3 Tensors

We diagrams tensors, which can be vertically and horizontally decomposed.

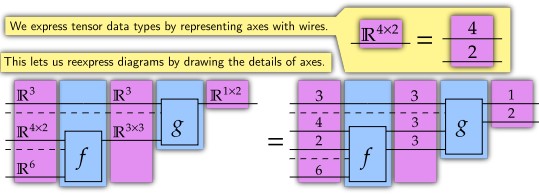

```
# This diagram shows a function h : 3, 4 2, 6 -> 1 2
constructed out of f: 4 2, 6 -> 3 3 and g: 3, 3 3 -> 1
2
# We use assertions and random outputs to represent
generic functions, and how diagrams relate to code.
T = torch.Tensor
def f(x0 : T, x1 : T):
    """ f: 4 2, 6 -> 3 3 """
    assert x0.size() == torch.Size([4,2])
    assert x1.size() == torch.Size([6])
    return torch.rand([3,3])
def g(x0 : T, x1: T):
    """ g: 3, 3 3 -> 1 2 """
    assert x0.size() == torch.Size([3])
    assert x1.size() == torch.Size([3, 3])
    return torch.rand([1,2])
def h(x0 : T, x1 : T, x2 : T):
    """ h: 3, 4 2, 6 -> 1 2"""
    assert x0.size() == torch.Size([3])
    assert x1.size() == torch.Size([4, 2])
    assert x2.size() == torch.Size([6])
    return g(x0, f(x1,x2))

h(torch.rand([3]), torch.rand([4, 2]), torch.rand([6]))
```

```
tensor([[0.6837, 0.6853]])
```

### 2.3.1 Indexes

Figure 8: Indexes

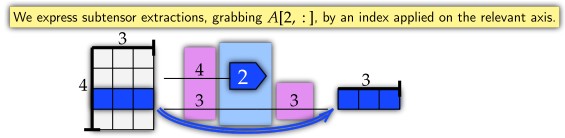

```
# Extracting a subtensor is a process we are familiar
with. Consider,
# A (4 3) tensor
table = torch.arange(0,12).view(4,3)
row = table[2,:]
row
```

```
tensor([6, 7, 8])
```

Figure 9: Subtensors

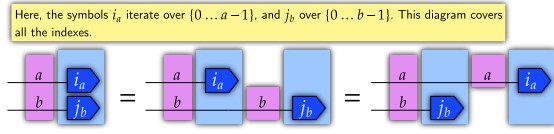

```
# Different orders of access give the same result.
# Set up a random (5 7) tensor
a, b = 5, 7
Xab = torch.rand([a] + [b])
# Show that all pairs of indexes give the same result
for ia, jb in itertools.product(range(a), range(b)):
    assert Xab[ia, jb] == Xab[ia, :][jb]
    assert Xab[ia, jb] == Xab[:, jb][ia]
```

### 2.3.2 Broadcasting

Figure 10: Broadcasting

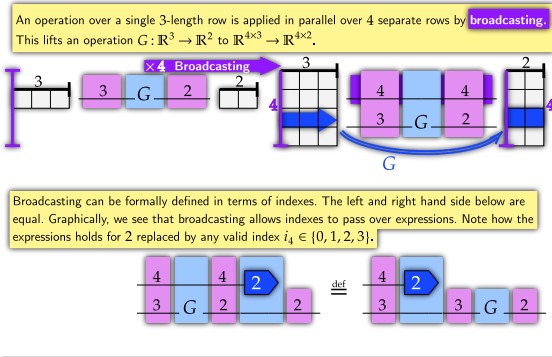

```
a, b, c, d = [3], [2], [4], [3]
T = torch.Tensor

# We have some function from a to b;
def G(Xa: T) -> T:
    """ G: a -> b """
    return sum(Xa**2) + torch.ones(b)

# We could bootstrap a definition of broadcasting,
# Note that we are using spaces to indicate tensoring.
# We will use commas for tupling, which is in line with
standard notation while writing code.
def Gc(Xac: T) -> T:
    """ G c : a c -> b c """
    Ybc = torch.zeros(b + c)
    for j in range(c[0]):
        Ybc[:,jc] = G(Xac[:,jc])
    return Ybc

# Or use a PyTorch command,
# G *: a * -> b *
Gs = torch.vmap(G, -1, -1)

# We feed a random input, and see whether applying an
index before or after
# gives the same result.
Xac = torch.rand(a + c)
for jc in range(c[0]):
    assert torch.allclose(G(Xac[:,jc]), Gc(Xac)[:,jc])
    assert torch.allclose(G(Xac[:,jc]), Gs(Xac)[:,jc])
```

```
# This shows how our definition of broadcasting lines
up with that used by PyTorch vmap.
```

Figure 11: Inner Broadcasting

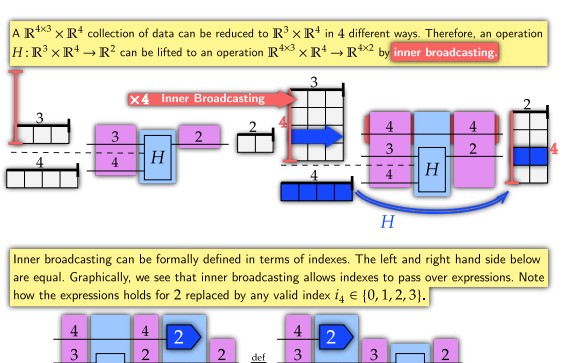

```
a, b, c, d = [3], [2], [4], [3]
T = torch.Tensor

# We have some function which can be inner broadcast,
def H(Xa: T, Xd: T) -> T:
    """ H: a, d -> b """
    return torch.sum(torch.sqrt(Xa**2)) +
torch.sum(torch.sqrt(Xd ** 2))  + torch.ones(b)

# We can bootstrap inner broadcasting,
def Hc0(Xca: T, Xd : T) -> T:
    """ c0 H: c a, d -> c d """
     # Recall that we defined a, b, c, d in [_] arrays.
    Ycb = torch.zeros(c + b)
    for ic in range(c[0]):
        Ycb[ic,  :] = H(Xca[ic, :], Xd)
    return Ycb

# But vmap offers a clear way of doing it,
# *0 H: * a, d -> * c
Hs0 = torch.vmap(H, (0, None), 0)

# We can show this satisfies Definition 2.14 by,
Xca = torch.rand(c + a)
Xd = torch.rand(d)
for ic in range(c[0]):
    assert torch.allclose(Hc0(Xca, Xd)[ic, :],
H(Xca[ic, :], Xd))
    assert torch.allclose(Hs0(Xca, Xd)[ic, :],
H(Xca[ic, :], Xd))
```

Figure 12 Elementwise operations

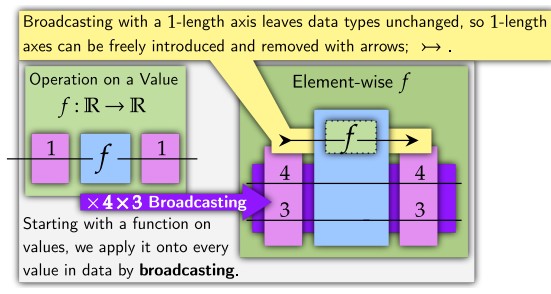

```
# Elementwise operations are implemented as usual ie
def f(x):
    "f : 1 -> 1"
    return x ** 2

# We broadcast an elementwise operation,
# f *: * -> *
fs = torch.vmap(f)

Xa = torch.rand(a)
for i in range(a[0]):
    # And see that it aligns with the index before =
index after framework.
    assert torch.allclose(f(Xa[i]), fs(Xa)[i])
    # But, elementwise operations are implied, so no
special implementation is needed.
    assert torch.allclose(f(Xa[i]), f(Xa)[i])
```

## 2.4 Linearity

### 2.4.2 Implementing Linearity and Common Operations

Figure 17: Multi-head Attention and Einsum

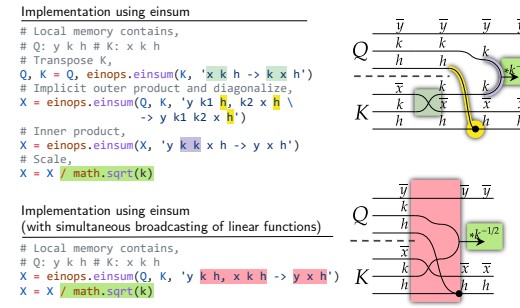

```
import math
import einops
x, y, k, h = 5, 3, 4, 2
Q = torch.rand([y, k, h])
K = torch.rand([x, k, h])

# Local memory contains,
# Q: y k h # K: x k h
# Outer products, transposes, inner products, and
# diagonalization reduce to einops expressions.
# Transpose K,
K = einops.einsum(K, 'x k h -> k x h')
# Outer product and diagonalize,
X = einops.einsum(Q, K, 'y k1 h, k2 x h -> y k1 k2 x
h')
# Inner product,
X = einops.einsum(X, 'y k k x h -> y x h')
# Scale,
X = X / math.sqrt(k)

Q = torch.rand([y, k, h])
K = torch.rand([x, k, h])

# Local memory contains,
# Q: y k h # K: x k h
X = einops.einsum(Q, K, 'y k h, x k h -> y x h')
X = X / math.sqrt(k)
```

### 2.4.3 Linear Algebra

Figure 18: Graphical Linear Algebra

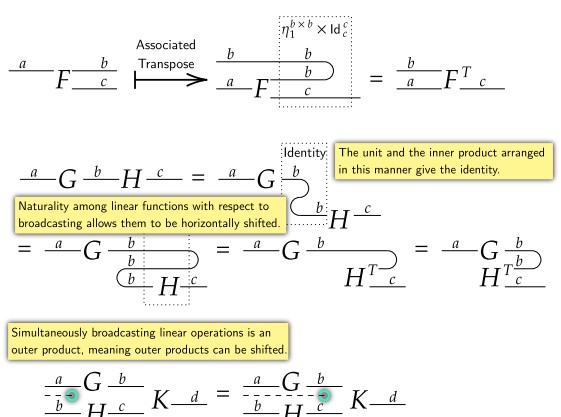

```
# We will do an exercise implementing some of these
equivalences.
# The reader can follow this exercise to get a better
sense of how linear functions can be implemented,
# and how different forms are equivalent.

a, b, c, d = [3], [4], [5], [3]

# We will be using this function *a lot*
es = einops.einsum

# F: a b c
F_matrix = torch.rand(a + b + c)

# As an exericse we will show that the linear map F: a
-> b c can be transposed in two ways.
# Either, we can broadcast, or take an outer product.
We will show these are the same.

# Transposing by broadcasting
#
def F_func(Xa: T):
    """ F: a -> b c """
    return es(Xa,F_matrix,'a,a b c->b c',)
# * F: * a -> * b c
F_broadcast = torch.vmap(F_func, 0, 0)

# We then reduce it, as in the diagram,
# b a -> b b c -> c
def F_broadcast_transpose(Xba: T):
    """ (b F) (.b c): b a -> c """
    Xbbc = F_broadcast(Xba)
    return es(Xbbc, 'b b c -> c')

# Transpoing by linearity
#
# We take the outer product of Id(b) and F, and follow
up with a inner product.
# This gives us,
F_outerproduct = es(torch.eye(b[0]), F_matrix,'b0 b1, a
b2 c->b0 b1 a b2 c',)
# Think of this as Id(b) F: b0 a -> b1 b2 c arranged
into an associated b0 b1 a b2 c tensor.
# We then take the inner product. This gives a (b a c)
matrix, which can be used for a (b a -> c) map.
F_linear_transpose = es(F_outerproduct,'b B a B c->b a
c',)

# We contend that these are the same.
#
Xba = torch.rand(b + a)
assert torch.allclose(
    F_broadcast_transpose(Xba),
    es(Xba,F_linear_transpose, 'b a, b a c -> c'))

# Furthermore, lets prove the unit-inner product
```

```
identity.
#
# The first step is an outer product with the unit,
outerUnit = lambda Xb: es(Xb, torch.eye(b[0]), 'b0, b1
b2 -> b0 b1 b2')
# The next is a inner product over the first two axes,
dotOuter = lambda Xbbb: es(Xbbb, 'b0 b0 b1 -> b1')
# Applying both of these *should* be the identity, and
hence leave any input unchanged.
Xb = torch.rand(b)
assert torch.allclose(
    Xb,
    dotOuter(outerUnit(Xb)))

# Therefore, we can confidently use the expressions in
Figure 18 to manipulate expressions.
```

## 3.1 Basic Multi-Layer Perceptron

Figure 19: Implementing a Basic Multi-Layer Perceptron

```
import torch.nn as nn
# Basic Image Recogniser
# This is a close copy of an introductory PyTorch
tutorial:
#
https://pytorch.org/tutorials/beginner/basics/buildmode
l_tutorial.html
class BasicImageRecogniser(nn.Module):
    def __init__(self):
        super().__init__()
        self.flatten = nn.Flatten()
        self.linear_relu_stack = nn.Sequential(
            nn.Linear(28*28, 512),
            nn.ReLU(),
            nn.Linear(512, 512),
            nn.ReLU(),
            nn.Linear(512, 10),
        )
    def forward(self, x):
        x = self.flatten(x)
        x = self.linear_relu_stack(x)
        y_pred = nn.Softmax(x)
        return y_pred

my_BasicImageRecogniser = BasicImageRecogniser()
my_BasicImageRecogniser.forward(torch.rand([1,28,28]))
```

```
Softmax(
  dim=tensor([[ 0.0150, -0.0301,  0.1395, -0.0558,
0.0024, -0.0613, -0.0163,  0.0134,
          0.0577, -0.0624]], grad_fn=
<AddmmBackward0>)
)
```

## 3.2 Neural Circuit Diagrams for the Transformer Architecture

Figure 20: Scaled Dot-Product Attention

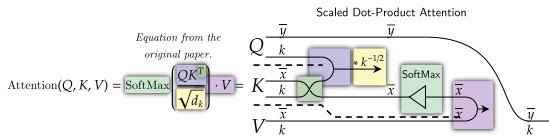

```
# Note, that we need to accomodate batches, hence the
... to capture additional axes.

# We can do the algorithm step by step,
def ScaledDotProductAttention(q: T, k: T, v: T) -> T:
    ''' yk, xk, xk -> yk '''
    klength = k.size()[-1]
    # Transpose
    k = einops.einsum(k,    '... x k -> ... k x')
    # Matrix Multiply / Inner Product
    x = einops.einsum(q, k, '... y k, ... k x -> ... y
x')
    # Scale
    x = x / math.sqrt(klength)
    # SoftMax
    x = torch.nn.Softmax(-1)(x)
    # Matrix Multiply / Inner Product
    x = einops.einsum(x, v, '... y x, ... x k -> ... y
k')
    return x

# Alternatively, we can simultaneously broadcast linear
functions.
def ScaledDotProductAttention(q: T, k: T, v: T) -> T:
    ''' yk, xk, xk -> yk '''
    klength = k.size()[-1]
    # Inner Product and Scale
    x = einops.einsum(q, k, '... y k, ... x k -> ... y
x')
    # Scale and SoftMax
    x = torch.nn.Softmax(-1)(x / math.sqrt(klength))
    # Final Inner Product
    x = einops.einsum(x, v, '... y x, ... x k -> ... y
k')
    return x
```

Figure 21: Multi-Head Attention

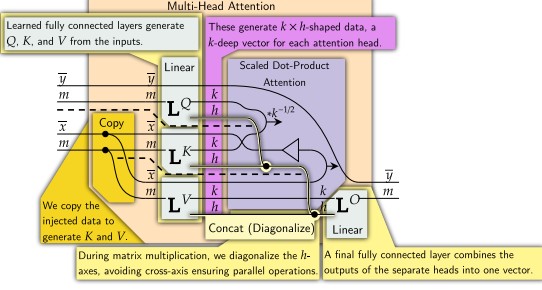

We will be implementing this algorithm. This shows us how we go from diagrams to implementations, and begins to give an idea of how organized diagrams leads to organized code.

```
def MultiHeadDotProductAttention(q: T, k: T, v: T) ->
T:
    ''' ykh, xkh, xkh -> ykh '''
    klength = k.size()[-2]
    x = einops.einsum(q, k, '... y k h, ... x k h ->
... y x h')
    x = torch.nn.Softmax(-2)(x / math.sqrt(klength))
    x = einops.einsum(x, v, '... y x h, ... x k h ->
```

```
... y k h')
    return x

# We implement this component as a neural network
model.
# This is necessary when there are bold, learned
components that need to be initialized.
class MultiHeadAttention(nn.Module):
    # Multi-Head attention has various settings, which
become variables
    # for the initializer.
    def __init__(self, m, k, h):
        super().__init__()
        self.m, self.k, self.h = m, k, h
        # Set up all the boldface, learned components
        # Note how they bind axes we want to split,
        # which we do later with einops.
        self.Lq = nn.Linear(m, k*h, False)
        self.Lk = nn.Linear(m, k*h, False)
        self.Lv = nn.Linear(m, k*h, False)
        self.Lo = nn.Linear(k*h, m, False)

    # We have endogenous data (Eym) and external /
injected data (Xxm)
    def forward(self, Eym, Xxm):
        """ y m, x m -> y m """
        # We first generate query, key, and value
vectors.
        # Linear layers are automatically broadcast.

        # However, the k and h axes are bound. We
define an unbinder to handle the outputs,
        unbind = lambda x: einops.rearrange(x, '... (k
h)->... k h', h=self.h)
        q = unbind(self.Lq(Eym))
        k = unbind(self.Lk(Xxm))
        v = unbind(self.Lv(Xxm))

        # We feed q, k, and v to standard Multi-Head
inner product Attention
        o = MultiHeadDotProductAttention(q, k, v)

        # Rebind to feed to the final learned layer,
        o = einops.rearrange(o, '... k h-> ... (k h)',
h=self.h)
        return self.Lo(o)

# Now we can run it on fake data;
y, x, m, jc, heads = [20], [22], [128], [16], 4
# Internal Data
Eym = torch.rand(y + m)
# External Data
Xxm = torch.rand(x + m)

mha = MultiHeadAttention(m[0],jc[0],heads)
assert list(mha.forward(Eym, Xxm).size()) == y + m
```

## 3.4 Computer Vision

Here, we really start to understand why splitting diagrams into ``fenced off'' blocks aids implementation. In addition to making diagrams easier to understand and patterns more clearn, blocks indicate how code can structured and organized.

Figure 26: Identity Residual Network

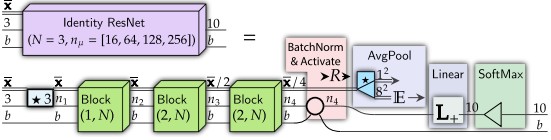

```python
# For Figure 26, every fenced off region is its own
module.

# Batch norm and then activate is a repeated motif,
class NormActivate(nn.Sequential):
    def __init__(self, nf, Norm=nn.BatchNorm2d,
Activation=nn.ReLU):
        super().__init__(Norm(nf), Activation())

def size_to_string(size):
    return " ".join(map(str,list(size)))

# The Identity ResNet block breaks down into a
manageable sequence of components.
class IdentityResNet(nn.Sequential):
    def __init__(self, N=3, n_mu=[16,64,128,256],
y=10):
        super().__init__(
            nn.Conv2d(3, n_mu[0], 3, padding=1),
            Block(1, N, n_mu[0], n_mu[1]),
            Block(2, N, n_mu[1], n_mu[2]),
            Block(2, N, n_mu[2], n_mu[3]),
            NormActivate(n_mu[3]),
            nn.AdaptiveAvgPool2d(1),
            nn.Flatten(),
            nn.Linear(n_mu[3], y),
            nn.Softmax(-1),
            )
```

The Block can be defined in a seperate model, keeping the code manageable and closely connected to the diagram.

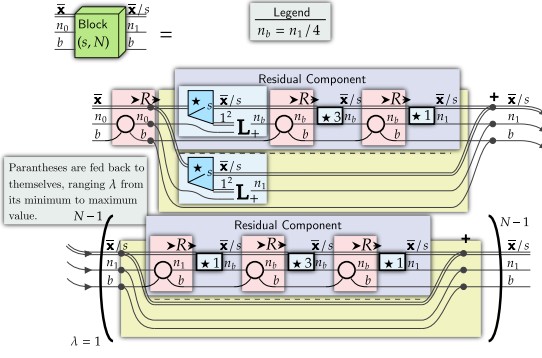

```python
# We then follow how diagrams define each ``block''
class Block(nn.Sequential):
    def __init__(self, s, N, n0, n1):
        """ n0 and n1 as inputs to the initializer are
implicit from having them in the domain and codomain in
the diagram. """
        nb = n1 // 4
        super().__init__(
            *[
            NormActivate(n0),
            ResidualConnection(
                nn.Sequential(
                    nn.Conv2d(n0, nb, 1, s),
                    NormActivate(nb),
                    nn.Conv2d(nb, nb, 3, padding=1),
                    NormActivate(nb),
                    nn.Conv2d(nb, n1, 1),
                ),
                nn.Conv2d(n0, n1, 1, s),
            )
            ] + [
            ResidualConnection(
                nn.Sequential(
                    NormActivate(n1),
                    nn.Conv2d(n1, nb, 1),
                    NormActivate(nb),
```

```python
                    nn.Conv2d(nb, nb, 3, padding=1),
                    NormActivate(nb),
                    nn.Conv2d(nb, n1, 1)
                ),
                )
            ] * N

            )
# Residual connections are a repeated pattern in the
diagram. So, we are motivated to encapsulate them
# as a seperate module.
class ResidualConnection(nn.Module):
    def __init__(self, mainline : nn.Module, connection
: nn.Module | None = None) -> None:
        super().__init__()
        self.main = mainline
        self.secondary = nn.Identity() if connection ==
None else connection
    def forward(self, x):
        return self.main(x) + self.secondary(x)
```

```python
# A standard image processing algorithm has inputs
shaped b c h w.
b, c, hw = [3], [3], [16, 16]

idresnet = IdentityResNet()
Xbchw = torch.rand(b + c + hw)

# And we see if the overall size is maintained,
assert list(idresnet.forward(Xbchw).size()) == b + [10]
```

The UNet is a more complicated algorithm than residual networks. The ``fenced off'' sections help keep our code organized. Diagrams streamline implementation, and helps keep code organized.

Figure 27: The UNet architecture

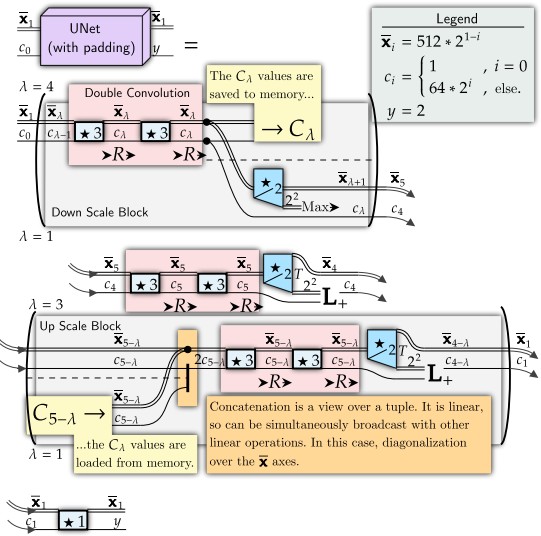

```python
# We notice that double convolution where the numbers
of channels change is a repeated motif.
# We denote the input with c0 and output with c1.
# This can also be done for subsequent members of an
iteration.
# When we go down an iteration eg. 5, 4, etc. we may
have the input be c1 and the output c0.
class DoubleConvolution(nn.Sequential):
    def __init__(self, c0, c1, Activation=nn.ReLU):
        super().__init__(
            nn.Conv2d(c0, c1, 3, padding=1),
```

```
            Activation(),
            nn.Conv2d(c0, c1, 3, padding=1),
            Activation(),
            )

# The model is specified for a very specific number of
layers,
# so we will not make it very flexible.
class UNet(nn.Module):
    def __init__(self, y=2):
        super().__init__()
        # Set up the channel sizes;
        c = [1 if i == 0 else 64 * 2 ** i for i in
range(6)]

        # Saving and loading from memory means we can
not use a single,
        # sequential chain.

        # Set up and initialize the components;
        self.DownScaleBlocks = [
            DownScaleBlock(c[i],c[i+1])
            for i in range(0,4)
        ] # Note how this imitates the lambda operators
in the diagram.
        self.middleDoubleConvolution =
DoubleConvolution(c[4], c[5])
        self.middleUpscale = nn.ConvTranspose2d(c[5],
c[4], 2, 2, 1)
        self.upScaleBlocks = [
            UpScaleBlock(c[5-i],c[4-i])
            for i in range(1,4)
        ]
        self.finalConvolution = nn.Conv2d(c[1], y)

    def forward(self, x):
        cLambdas = []
        for dsb in self.DownScaleBlocks:
            x, cLambda = dsb(x)
            cLambdas.append(cLambda)
        x = self.middleDoubleConvolution(x)
        x = self.middleUpscale(x)
        for usb in self.upScaleBlocks:
            cLambda = cLambdas.pop()
            x = usb(x, cLambda)
        x = self.finalConvolution(x)

class DownScaleBlock(nn.Module):
    def __init__(self, c0, c1) -> None:
        super().__init__()
        self.doubleConvolution = DoubleConvolution(c0,
c1)
        self.downScaler = nn.MaxPool2d(2, 2, 1)
    def forward(self, x):
        cLambda = self.doubleConvolution(x)
        x = self.downScaler(cLambda)
        return x, cLambda

class UpScaleBlock(nn.Module):
    def __init__(self, c1, c0) -> None:
        super().__init__()
        self.doubleConvolution =
DoubleConvolution(2*c1, c1)
        self.upScaler = nn.ConvTranspose2d(c1,c0,2,2,1)
    def forward(self, x, cLambda):
        # Concatenation occurs over the C channel axis
(dim=1)
        x = torch.concat(x, cLambda, 1)
        x = self.doubleConvolution(x)
        x = self.upScaler(x)
        return x
```

## 3.5 Vision Transformer

We adapt our code for Multi-Head Attention to apply it to the vision case. This is a good exercise in how neural circuit diagrams allow code to be easily adapted for new modalities.

Figure 28: Visual Attention

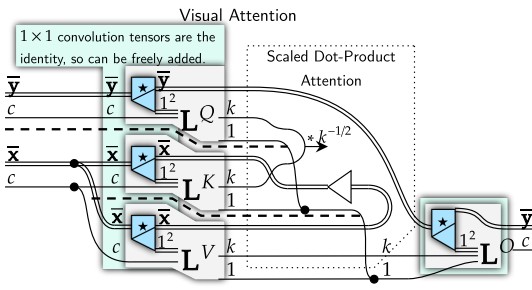

```
class VisualAttention(nn.Module):
    def __init__(self, c, k, heads = 1, kernel = 1,
stride = 1):
        super().__init__()

        # w gives the kernel size, which we make
adjustable.
        self.c, self.k, self.h, self.w = c, k, heads,
kernel
        # Set up all the boldface, learned components
        # Note how standard components may not have
axes bound in
        # the same way as diagrams. This requires us to
rearrange
        # using the einops package.

        # The learned layers form convolutions
        self.Cq = nn.Conv2d(c, k * heads, kernel,
stride)
        self.Ck = nn.Conv2d(c, k * heads, kernel,
stride)
        self.Cv = nn.Conv2d(c, k * heads, kernel,
stride)
        self.Co = nn.ConvTranspose2d(
                        k * heads, c, kernel,
stride)

    # Defined previously, closely follows the diagram.
    def MultiHeadDotProductAttention(self, q: T, k: T,
v: T) -> T:
        ''' ykh, xkh, xkh -> ykh '''
        klength = k.size()[-2]
        x = einops.einsum(q, k, '... y k h, ... x k h -
> ... y x h')
        x = torch.nn.Softmax(-2)(x /
math.sqrt(klength))
        x = einops.einsum(x, v, '... y x h, ... x k h -
> ... y k h')
        return x

    # We have endogenous data (EYc) and external /
injected data (XXc)
    def forward(self, EcY, XcX):
        """ cY, cX -> cY
        The visual attention algorithm. Injects
information from Xc into Yc. """
        # query, key, and value vectors.
        # We unbind the k h axes which were produced by
the convolutions, and feed them
        # in the normal manner to
MultiHeadDotProductAttention.
        unbind = lambda x: einops.rearrange(x, 'N (k h)
H W -> N (H W) k h', h=self.h)
        # Save size to recover it later
        q = self.Cq(EcY)
        W = q.size()[-1]
```

```python
        # By appropriately managing the axes, minimal
changes to our previous code
        # is necessary.
        q = unbind(q)
        k = unbind(self.Ck(XcX))
        v = unbind(self.Cv(XcX))
        o = self.MultiHeadDotProductAttention(q, k, v)

        # Rebind to feed to the transposed convolution
layer.
        o = einops.rearrange(o, 'N (H W) k h -> N (k h)
H W',
                             h=self.h, W=W)
        return self.Co(o)

# Single batch element,
b = [1]
Y, X, c, k = [16, 16], [16, 16], [33], 8
# The additional configurations,
heads, kernel, stride = 4, 3, 3

# Internal Data,
EYc = torch.rand(b + c + Y)
# External Data,
XXc = torch.rand(b + c + X)

# We can now run the algorithm,
visualAttention = VisualAttention(c[0], k, heads,
kernel, stride)

# Interestingly, the height/width reduces by 1 for
stride
# values above 1. Otherwise, it stays the same.
visualAttention.forward(EYc, XXc).size()
```

```
torch.Size([1, 33, 15, 15])
```

# Appendix

```python
# A container to track the size of modules,
# Replace a module definition eg.
# > self.Cq = nn.Conv2d(c, k * heads, kernel, stride)
# With;
# > self.Cq = Tracker(nn.Conv2d(c, k * heads, kernel,
stride), "Query convolution")
# And the input / output sizes (to check diagrams) will
be printed.
class Tracker(nn.Module):
    def __init__(self, module: nn.Module, name : str =
""):
        super().__init__()
        self.module = module
        if name:
            self.name = name
        else:
            self.name = self.module._get_name()
    def forward(self, x):
        x_size = size_to_string(x.size())
        x = self.module.forward(x)
        y_size = size_to_string(x.size())
        print(f"{self.name}: \t {x_size} -> {y_size}")
        return x
```

