# OpenReview forum: "Neural Circuit Diagrams: Robust Diagrams for the Communication, Implementation, and Analysis of Deep Learning Architectures"
_TMLR — Accepted by TMLR_

### Review · Reviewer_82Vd · 2023-07-11

**Summary Of Contributions:**

This paper presents a diagramming method for communicating deep learning models, focusing on transformers.  Notations are described for tensor representation, combination, and function application.  After examples describing MLP and multi-head attention, the paper culminates in describing a transformer model.

**Audience:**

No

**Claims And Evidence:**

No

**Requested Changes:**

Improvements in at least one of (1) and/or (2) below are critical, in my view (and preferably both).  Additionally, (3) would further strengthen the work.

1. Research into the practical usefulness and benefits of these diagrams would help bring a level of rigor, for example user studies or questionnaires.  Maybe a study where a user needs to code up an implementation based on a description with vs without the new diagram, for example?  Or a questionnaire asking readers to answer questions about specifics with vs without this diagram?  Diagram vs equations?

2. Likewise, pairing this with an automated tool that can generate these diagrams, perhaps based on a pytorch definition, would be useful in encouraging usage and potential adoption.

3. The diagrams are applied exclusively to sequence transformers (and a small MLP), as case studies.  However, other models and domains can have different details that are important to get across.  Object detectors, for example, are principally concerned with object scale and alignment, some of which can be shown with shape dimensions (scale), but others (alignments, anchors) are not as clear.



**Strengths And Weaknesses:**

I appreciate the need for improving diagrams, and even standardizing to a greater degree.  The notation and techniques suggested here offer some nice ways looking at and communicating operations, particularly the analogy between line connections and Einstein summation notation, and improved clarity of which operations apply to which dimensions.  Certainly, a reader can get some good ideas for drawing effective specifications.

However, I think this currently lacks the empirical rigor and/or interesting enough technical contribution to meet TMLR criteria.  The current paper presents a system of diagramming, but no evidence to support its improvement over a combination of ad-hoc diagrams, equations and descriptions (user studies or questionnaires could supply this, for example).  Similarly, a script or tool that can automatically generate these figures (or a first draft that can be edited) would go a long way in encouraging adoption (or adoption of pieces), as researchers could view more different models to get a feel for how the diagrams work (and without this, I wouldn't expect authors to be able to get all the details right).


---

Putting forward these diagrams makes it all too tempting to provide nitpicks on the diagrams and styles themselves.  I don't think it's appropriate to base this review on my own response to these particular diagrams, and tried to give more substantive feedback first, both above and in the "requested changes" section below.  Still, I've listed many of my observations, which I hope can also be helpful:

* the colored boxes are a little too busy, and group together data that is separate; dotted lines separate tensors, but the pink boxes are more apparent and look like the larger grouping.  is it possible to write these in black-and-white or grayscale, with color optional addition?

* representing dimensions with different lines makes a lot of sense for some operations, particularly multi-dim transposes and multiplications, but gets in the way for others, e.g. copy operation (what if only the x dim is copied in fig 11 but not m, what does that even mean?)

* These diagrams are very exacting, with small details having large differences (the dashed lines indicating separate tensors is one of the largest here).  As such, I don't expect many researchers or engineers would include all details based purely on spec, leading again to ad-hoc/freeform diagrams, though with perhaps a few new useful notations.  That strikes me as similar to UML, which is scarcely followed to the letter.

* the scale by 1/sqrt(k) in fig 10 in particular isn't very intuitive to me.  The arced connection for matrix mul makes some sense after getting used to it (it combines the two dims, similar to einstein summation notation, as pointed out), but the arrow doesn't really go anywhere to connect to anything.  Scaling could just as easily be applied over any of these dimensions (y, k, x).

* reshaping is a fundamental operation I didn't see covered.  It would be convenient to refer to passing multi-dim tensors with a single line, and split into multiple dims only when needed.

* taking a page from circuit diagrams, what about using cross-hatch with the dimension to denote shapes?

* softmax arrow notation is nice, though I think I've seen right-pointing arrow used for this in the past more often, suggesting comparison into (soft) 1-hot outputs


* binding diagram is almost the same as copying (no dotted line) but with much different effect.  also keeping in mind addition is the grad of copy, it would make sense for the copy diagram flipped horizontally to perform a sum, but the "bind" operation doesn't do that (getting diagonal values of a matrix also not very common)

---

> ### Author Response · Authors · 2023-07-12
> **Thank you for your review**
>
> Thank you for the review; we really appreciate it and find it very helpful. They are all valuable suggestions. We were indecisive about whether to include additional technical content or keep this paper short. Your comments are valuable in that they indicate that the first approach would be better. We hoped that an accessible presentation of our notation would encourage follow-up work concerning the points you mentioned - forming the basis for user studies and automated tools.
>
> We feel like a comparison to Named Tensor Notation - another paper published by TMLR - is apt. Our goal is very similar, but we decided to differ our presentation to be more accessible and to be around the twelve-page guide for a regular submission. Your comments are valuable because they show us that a longer, broader paper might be necessary to convey our ideas properly.
>
> Concerning point (1), a user study as a follow-up to compare different notational systems would be fascinating, and we were hoping could be follow-up research after first establishing the technical basis of our work. Named Tensor Notation did not include a user study. Instead, it justified its claims by presenting its notation for many systems. In contrast, we covered a specific case in depth (transformers) and went into detail about the shortfalls of the original presentation. However, we would happily extend our work by covering the same breadth of cases as Named Tensor Notation. We are confident in our notation's ability to readily generate accurate diagrams for many cases, so we would be happy to extend the breadth of cases we study. By bringing the breadth of our work in line with Named Tensor Notation, would this address the practical usefulness and benefits concern of point (1)?
>
> For point (2), an automated tool is a great idea. Covering more cases in the paper would begin addressing the concerns of giving researchers a better feel for these diagrams. However, an automated tool can also be made to generate diagrams for many cases automatically. PyTorch's gradient graph should make this a very doable development.
>
> For point (3), we are unfamiliar with object detectors. The point about covering different domains and case studies is very fair. This point makes us think about convolutional layers and how they are covered particularly well by neural circuit diagrams. Their close relationship to pooling operations becomes clear and concepts such as stride and dilation are demystified in a neural circuit diagram presentation. We could present that analysis to broaden the case studies we cover and show how our diagrams have particular merit in presenting important concepts in machine learning.
>
> However, even addressing these points, there seems to still be concern about our presentation's technical depth and merit. This paper aimed to present neural circuit diagrams in an accessible form. We have additional results about analyzing the gradients of diagrams and a category theory basis for their construction. This category theory perspective allows for analyzing diagrams as Markov categories, allowing information theory and statistical analysis of diagrams to be conducted, giving insights into the behavior of data within networks. The category theory perspective also shows why it is necessary to have two ways of combining axes - coupling (tensor products) and collecting (cartesian products) - and how this innovation overcomes the limitations of previous graphical methods, which, as Named Tensor Notation points out, have been limited to purely linear systems.
>
> These results have technical depth and applicability, but we were concerned about the relevance of including them in the first paper on neural circuit diagrams. Would including some of these results in this paper address your concerns about supported claims and evidence and audience relevance for this work? Perhaps we should only include the gradient analysis, therefore having the same technical breadth and depth as Named Tensor Notation without going into the specifics of category theory?
>
> We will also address the specifics of your minor observations, restructuring the work to have specific sections on implementing diagrams in PyTorch and einops operations, including binding/diagonalizing a matrix.
>
> Thank you again for your review; we have started incorporating our other results into the paper to address the concerns about justified claims and evidence and to bring our work more in line with previous submissions to TMLR.

---

> ### Author Response · Authors · 2023-09-24
> **Updates on latest revision**
>
> The latest revision is significantly different from the version you reviewed initially. It addresses the points you raised and, we feel, is in a strong position to be accepted to TMLR.
>
> Our work aligns with many of the [enumerated scope points of TMLR](https://jmlr.org/tmlr/editorial-policies.html). Two, however, clearly stand out:
> - development of new analytical frameworks that advance theoretical studies of practical learning methods;
> - new approaches for analysis, visualization, and understanding of artificial or biological learning systems.
>
> Section 1 covers the case for diagrams, particularly the sequential tuple-tensor diagrams we employ. Previous successes are surveyed, and the critical technical problem to be solved - reconciling tuple and tensor graphical representations - is identified. We then cover the philosophy of our approach, expressing how we believe diagrams should deal with various levels of abstraction and whether contributing a diagramming standard in the first place is worthwhile.
>
> Having identified this issue and its relevance to the deep learning community, we lay out neural circuit diagrams with diagrams of gradually increasing complexity. Compared to the original version, we have the luxury of many more pages, and our current presentation is more comprehensive, motivating understanding and adoption.
>
> Then, we diagram a host of architectures. Along the way, we identify the advantages of neural circuit diagrams over current approaches. Additionally, we accompany our diagrams with code, showing the close relationship between our diagrams and implementation, further motivating relevance to an audience of deep learning researchers. This section also provides evidence for our claim that our diagrams offer a significant improvement over current methods.
>
> We end with an analysis of differentiation and backpropagation. This section puts our technical contribution on par and indeed deeper than [previous TMLR works](https://openreview.net/pdf?id=hVT7SHlilx), in addition to further showing and evidencing the value of neural circuit diagrams to machine learning researchers.
>
> Overall, the current version identifies a critical problem in deep learning research, contributes a solution, and evidences this solution through several case studies with accompanying code. The claims are evidenced throughout the work, and relevance to the audience is emphasized. Technical depth is offered during the introduction of diagrams, the diagramming of various architectures, and the careful analysis of differentiation and backpropagation.

---

> > ### Comment · Reviewer_82Vd · 2023-10-05
> > **new revision**
> >
> > Thanks for the new revision and comments.  I've only been able to briefly skim through so far, but based on that it looks much improved, with more grounded correspondences between the diagrams and model implementation code.  I'm reading through in more depth now and hope to provide a more full response by early next week.

---

> ### Comment · Reviewer_82Vd · 2023-10-13
> **new revision comments**
>
> I've read the latest revision in more detail.  I think it is much improved over both previous versions.  In particular, it is much more concrete in its correspondences between diagrams and operations, and provides examples of multiple architectures (now mlp, transformer, convnet, u-net and vit).  There has been a back-and-forth on the level of mathematical depth, to make the diagrams concretely meaningful on the one hand but still accessible on the other --- IMO the latest revision strikes a decent balance, though perhaps still a bit on the overly detailed side.
>
> I like also how it lays out the building blocks more step-by-step, using terminology familiar to the those in the deep learning field (e.g. "tuple" instead of "collection", "tensor" instead of "coupling").  Though this and the new architectures do make the new revision quite long at 25 pages, much of this space I think is appropriate due to the need for many diagrams introducing the various components and operations (in addition, I don't think TMLR has any strict page limit).
>
> While many of the building blocks are now more concrete, there are a few important ones that I feel could be more explicitly explained; somehow it seems a few of the following basics got lost in going from introductory diagrams to the more advanced ones.  Many of these are around how the diagrams denote linear operations, which may be part of the string diagrams they build upon, but I'm not familiar with those:
>
> * sec 2.4.2, 2.4.3:  It looks like inner product is the curved line here, but that isn't explicit anywhere in these sections.  It's used for QK' but that is in the middle of a larger example.  I think it would make sense to define the inner product curved connection notation separately using a simple example.
>
> * 2.4.3: transposing a multilinear function may not make sense to many in this audience  (I struggled a bit to see what was happening here, even with a bit of familiarity).  I'd expect most would be very familiar with transpose of linear functions, and use the term "transpose" for reshaping tensor axes, but not necessarily think of them as multilinear functions.  In addition, the $F^T$ in this example is $F^T : [b, a] \rightarrow [c]$, whereas in numpy, .T reverses the axes, so that for a tensor shape [c,b,a], $F : [a] \rightarrow [c, b]$ gives (depending on interpretation) either $F.T : [c] \rightarrow [a, b]$ or $F.T : [b, c] \rightarrow [a]$, neither of which are this diagram.  None of np.dot, np.matmul or np.tensordot seem to have a clear correspondence along with the transposes, either.  According to the diagram, it looks like it would peform $F^T(x_a, x_b) = (x_b \cdot y_b, y_c) = (x_b \cdot y_b) y_c$, where $(y_b, y_c) = F(x_a)$.  Is this correct (and if so, maybe better to state this, and/or use a more matrix-like example)?
>   (Note I'm not trying to use numpy here as a definitive source of any definition, but rather as an example of what might be expected by those familiar with this or similar libraries like torch for these terms and notations.)
>
> * 3.3:  I don't understand the diagram for transposed convolution.  What does it mean for y to have that roundabout wire to the right of the convolution op, and why is x still on the right?  This is similar to my confusion about transposes in sec 2.4.3 above, but even less clear in this convolutoins case.
>
> * Fig 33:  it was difficult to understand the meaning of the output a wire connecting to the input of DF --- in the past, for right-facing curved connection this was inner product.  the caption indicates that this "broadcasts" over a dimension --- which sounds like conceptually a loop over each dimension of a, calculating DF with each unit basis element.  But I don't think this was explained clearly.  In general, I don't see much need for the forwards derivative here --- why not skip straight to what is called the reverse derivative, which is more familiar to those familiar with backprop already? (the relationship to DF could go in an appendix).
>
>
> Additional minor observations and questions included in next comment.

---

> > ### Comment · Reviewer_82Vd · 2023-10-13
> > **additional feedback**
> >
> > Additional minor observations and questions:
> >
> > * p.5:  "It is outside the scope of this work to generalize diagrams"  - the setting is already quite general, so this phrase appears to down-play the scope of its own work.  could be tweaked to e.g. "Generalizing our diagrams to include architectures beyond this already extensive setting is beyond the scope of this work, though exciting avenue for future re
> > asarch on the topic.
> >
> > * indexes:  ket notation is nice, but IMO might make more sense to introduce using index lists/sets instead of single indices, since an output wire from the ket could be the index selection (aka "gather").
> > ```
> > N --------| inds >------ I
> > D ---------------------- N    =   X[inds, :]  where X.shape == (N,D), inds.shape == (I,)
> > ```
> > then for single indices, this wire can be dropped (implicit "squeeze")
> >
> >
> > * p.9  some chars maybe not rendering correctly for me, I see "This means a function [blank square] -> R^a" and "if two functions f,g : R^[crossed-out double-hatch] -> R^3"
> >
> >
> > * I understand the diagrams for broadcasting (they handle broadcasts pretty naturally), but the text in this section seems cumbersome
> >
> > * Fig 27 resnet:  I don't know what the circles that take (n,b) dims do --- were these described before?  I think they're batchnorms.  Other than that I think this is a nice diagram that shows what is happening well.
> >
> > * Fig 29 vit:  says, "1x1 convolution tensors are the identity, so can be freely added" --- I'm not sure what this means, as they are definitely not the identity matrix.  I think it could mean that they don't affect the shapes, which is true in the case of d x d mappings.  But it looks like something stronger might be intended?

---

> > > ### Author Response · Authors · 2023-10-19
> > > **Thank you for your comments**
> > >
> > > Thank you for taking the time to read our paper and provide in depth comments. We are glad you feel the current version looks much improved.
> > >
> > > We've uploaded a revision that addresses your concerns, but less us know if you have any additional comments.
> > >
> > > ### Important points
> > > - Section 2.4.2 mentions that inner products are represented with cups, which is explicitly shown in Figures 18 and 19.
> > > - We have rewritten Section 2.4.3 to make the idea of transposing the associated tensor clearer. We have included:
> > > >	These rearrangements can transpose specific axes. A linear operation $\mathbb{R}^{a\times b}\rightarrow \mathbb{R}^c$ has an associated $\mathbb{R}^{a\times b\times c}$ tensor. This tensor can be associated with various linear operations, such as $\mathbb{R}^{b\times a} \rightarrow \mathbb{R}^c$. These different forms are often of interest to us, as they can efficiently implement the reverse of operations (see Figure 26, 31). To extract these rearrangements, we can selectively apply units and the inner product to reorient the direction of wires for linear operations.
> > >
> > > This also better explains transposed convolution.
> > >
> > > ### Analyzing Derivatives
> > > Looping over each basis element is exactly what the left-facing cup $\subset$ is doing, while the right-facing cup $\supset$ multiplies the derivative by the target derivative of $\delta \mathcal{L}$. This returns the direction which changes $X$ so that $\mathcal{L}$ decreases. We have added comments to better explain this, and tweaked the diagram.
> > >
> > > We go through forward derivatives to show analytical tools available for neural circuit diagrams. Forward derivatives have a functor (composition preserving map on functions) that easily represents the chain rule. The functor for reverse derivatives is more complicated, requiring data to flow in two directions.
> > >
> > > There is also utility in using neural circuit diagrams to derive the computational complexity of forward derivatives then compare this to using reverse derivatives.
> > >
> > > ### Additional Minor Observations
> > > - We have changed the line in the introduction not to downplay the scope of the work.
> > > - Length 1-axes can be freely introduced and removed ("squeeze"). If you want, you could diagram a redundant length 1-axes. This could be used when interacting with [PyTorch](https://pytorch.org/docs/stable/notes/broadcasting.html) broadcasting semantics, or as in Figure 29 (visual attention) for $1\times 1$-kernels and to hint at multi-head visual attention.
> > > - We have fixed the rendering issue.
> > > - For Fig 27 ResNet, we have added that the pink blocks are batchnorm and activate blocks.
> > > - We've clarified the point about "being the identity." This refers to a $1\times 1$ convolution being the identity map, $\text{Conv}(v,[1]) = v$. This is why we can freely introduce it.
> > >
> > > If you have any further comments, we would be happy to hear them and make any necessary changes.

---

> > > > ### Comment · Reviewer_82Vd · 2023-10-22
> > > > **re: comments**
> > > >
> > > > Thanks, that clears up most of these smaller points.  I still think a dedicated figure and description of right-facing cup for inner product would be helpful, as its use in Figs 17,18 is in the middle of a lot of other things going on.

---

> > > > > ### Author Response · Authors · 2023-10-22
> > > > > **Additional figure (Figure 15) added to explicitly introduce the inner product.**
> > > > >
> > > > > We have uploaded a revision that adds Figure 15 which explicitly explains inner broadcasting and the inner product. This is a dedicated figure to introduce the right-facing cup. This opportunity is used to better explain inner broadcasting, and to show how matrix multiplication is an emergent property of the inner product and inner broadcasting.
> > > > >
> > > > > If you have additional comments, let us know and we can make the necessary changes.

---

### Review · Reviewer_XvGP · 2023-08-17

**Summary Of Contributions:**

This paper proposes Neural Circuit Diagrams as a way of visualizing deep learning architectures in detail. It revolves around shaped data whose axes are combined via coupling or collecting. Morphisms, having fixed input and output shapes act as operations on the data.

The paper proceeds with several examples of fundamental data operations (Figures 1–6). Afterward, a Neural Circuit Diagram of an MLP is shown (Figure 8), followed by diagrams of scaled-dot-product attention, multi-head attention, and eventually a transformer (Figures 10–12). Along the way, a few standard morphisms are introduced, such as for transposing, binding, and copying (Figure 7).


**Audience:**

Yes

**Claims And Evidence:**

Yes

**Requested Changes:**

I would like to see a lot more standardization and guidelines included for creating diagrams for many of the standard architectures. Examples of vision architectures like U-Nets or Inception nets, and recurrent nets like LSTMs would be particularly informative. Other examples focusing on handling multiple modalities (eg vision and text), or cases where processing proceeds differently along various input dimensions (eg. factorizing attention spatially and across time, as is commonly done in video generative models).

As I mentioned before, to be able to judge the potential impact of this work, it would be good to be more specific about the intended target audience and designer of such diagrams. It would also be good to clarify how it can be ensured that multiple diagrams of the same model end up being the same, since this would overall benefit adoption. Finally, commenting on how these diagrams complement annotated (pseudo-)code would also be useful, especially since readers may generally already be more familiar with code.

**Strengths And Weaknesses:**

**Strengths**

I strongly agree with the premise of this work, which is that deep learning papers and their diagrams omit too many details, which makes it difficult to reproduce them. This limits the overall pace at which advances can be incorporated.

The proposed neural circuit diagram look great from an artistic standpoint (as far as I am able to judge this). The bright colors are particularly appealing, and I can imagine how (once a certain degree of familiarity has been obtained) these diagrams could be useful when no code is provided.

**Weaknesses**

Generally, this paper is a bit unusual in the kind of contribution it makes, and I find it difficult to review. A framework is proposed for diagramming deep learning architectures, yet only few guidelines and standard notations are presented for creating these. For example, after reading the paper I am not a lot wiser about drawing these diagrams for other architectures like UNets, LSTMs, or deep CNNs like Inception Networks. Because of the lack of standardization, it feels that different people creating such diagrams would end up with different drawings, which perhaps defeats the purpose a little.

To highlight some broader issues in particular:

* Before reading the paper in detail, I was not able to understand the diagrams very well. I initially took some time to study Figure 12, and I honestly found it quite difficult to make sense of all the details. Operations like tensor products, softmax axis normalization, and other symbols for drop-out are not intuitive. Similarly, the distinction between lines for coupling and collection is difficult to keep track off. Perhaps these diagrams are not meant to be intuitive at a glance, and they behave more like code or pseudo-code whose language needs to be understood first. However, in that case, having annotated code seems more useful. I note that this could also be a lack of familiarity with these kind of visualizations that can improve over time.
* Few guidelines are provided for creating these kind of diagrams, and the level of detail used for various components of a models seems up to the user. For example, in the Transformer figure (Fig 12), the masking / dropout and normalization operations are presented in relatively little detail compared to the matrix multiplications. This is partially resolved via the text boxes, which almost act like a legend, though limited guidance is provided for structuring these. Because of this, it feels like there is no standard neural circuit diagram even for the Transformer, and different researchers may end up with different diagrams for the same thing. Indeed, after reading the paper in detail I don’t feel very confident in my own ability to create similar diagrams for other architectures (like models in my own papers). This could well be an issue with my own artistic abilities, yet it makes it difficult to judge the significance of this kind of diagramming. At the moment, I don’t see much benefit in adopting this kind of diagramming compared to sharing code, and I would not expect this approach to be well adopted in the community.

I would ask the authors, who these diagrams are meant for, in what situation they are expected to be used, and who the intended creates of such diagrams are. This would help determine the impact of this contribution and whether it is reasonable to expect adoption.

---

> ### Author Response · Authors · 2023-08-22
> **Thank you for your review**
>
> We are glad you enjoyed the artistic style we used! We consider the ability to accommodate heavy annotation (or no annotation) to be one of the advantages of neural circuit diagrams.
>
> We have revised the paper with your comments in mind. It has become unavoidably longer, but we feel the added content adds value for machine learning researchers.
> ## Weaknesses
> 1) **General Comments**
> 	1) **Difficulty to Review:** The original paper was atypical in presenting theory with one key result, the transformer architecture. To present further evidence, we needed more pages. The revision is longer, and we have more carefully laid out the introduction, discussing related works and why our approach is necessary. We have deepened our technical contributions by employing category theory. The results section now covers a host of architectures, as well as an exploration of backpropagation. The revision is easier to review. We hope this resolves the issue of determining whether the paper fits into existing research, and whether the claims it makes are evidenced and of interest to the machine learning community.
> 	2) **General Guidelines:** As we are introducing these diagrams, we can only cover so many cases. However, it is fair to say that more than one architecture and an exploration of computer vision is necessary to independently deploy these diagrams. Therefore, we covered many additional cases in our latest revision. This includes:
> 		- Convolution; with the utility of neural circuit diagrams evidenced in diagramming stride, dilation, and transposed convolutions.
> 		- The U-Net Architecture; notable as it forms the basis of complex generative image models.
> 		- The Identity ResNet; notable as common implementations of ResNet miss key architectural features, further evidencing imprecise communication in the community.
> 		- The Vision Transformer; notable as it shows the utility of neural circuit diagrams in comparing and innovating on existing architectures, and how multiple modalities can be similarly treated.
> 	3) **Standardization:** In standardizing diagrams, we are more interested in a common, precise graphical language, rather than having each team's diagrams look the same. We want diagrams to be expressive, and tailored to specific uses, as long as they are clear and precise.
> 2) **Point 1 - Unintuitive at First Glance**: We acknowledge there is a learning curve. However, a single learning curve for neural circuit diagrams is far more preferable than every single deep learning paper having a learning curve for its ad-hoc diagrams, which is currently the case. A small legend and some annotations are far better than a diagram that is impossible to dissect. The current revision of the paper has a higher learning curve than the original version. However, we feel this is appropriate as future work can broadly disseminate neural circuit diagrams, and the current revision covers valuable technical depth.
> 3) **Point 2 - Level of Detail and Using Code Instead** Annotated code, or improved symbolic expressions, are valuable. However, the overarching structure of architectures, the large data flows and the composition of many components, still needs to be expressed. This is best done visually. However, without a precise graphical language the diagrams inevitably included with papers will be difficult to understand. (See: Section 1.3, paragraph 1). Our revised introduction addresses this point (Section 1.1). Briefly, only sharing code creates a dependency on a specific framework, places a burden on the reader, obfuscates developments, and leads to weaker scientific results.
>
> ### Requested Changes
> The current revision addresses the concerns about standardization and guidelines. Rather than standardization specifically, we hope to encourage precision through a common graphical language (see above, 1.3). The technical depth is greatly increased, offering guidelines for other teams to diagram architectures and new components, confident that the diagrams they produce have precise mathematical meaning and clear implementations. The revision includes many more architectures (see above, 1.2). You may be particularly interested in Section 3.5 of the revised version, which discusses the advantages of neural circuit diagrams when mixing modalities.
>
> We have also addressed concerns about identifying the target audience. Neural circuit diagrams hope to be broadly used, and therefore the paper must be restricted to discuss their relevance for a particular group. We focus on machine learning researchers, discuss why neural circuit diagrams are relevant for them, and use a level of mathematics that should be accessible by the community. The key results focus on how neural circuit diagrams improve foundational understanding. As a result, the revision should be easier to judge (see above, 1.1).

---

> > ### Comment · Reviewer_XvGP · 2023-09-24
> >
> > Thank you for addressing my concerns. I appreciate including the technical details, which make the underlying technical contribution of these visualizations quite clear. Because of this, my previous concern as to whether these diagrams will likely be adopted by the community (and about which we can only speculate) is of lesser importance. I also appreciate the additional examples for standard architectures, which were missing before. Overall my recommendation will be an accept.
> >
> > From the other reviews, I can see that other reviews are slightly concerned that paper has become too focused on technical material (which perhaps is in part my fault). I think that perhaps a middle ground regarding technical details can be to move some of this to the back of the paper as "additional reading", though I remain of the opinion that it is important that these details are included as they provide needed technical grounding for the proposed diagrams.

---

> > > ### Author Response · Authors · 2023-09-24
> > > **Thank you for your comment**
> > >
> > > We appreciate your support and have found your advice valuable. The new version should be more accessible. The new version introduces the technical foundations of diagrams without category theory but with equal depth. A deeper analysis of differentiation and backpropagation has been added, further proving the technical contribution of these diagrams. We feel the current paper is in a strong state to support future research while being specialized for an audience of deep learning researchers.

---

### Review · Reviewer_wTkA · 2023-09-07

**Summary Of Contributions:**

This paper proposes a novel way of presenting tensor transformations as commonly used in deep learning architectures. The so called neural circuit diagrams are effectively a well-defined graphical language that can exactly describe deep networks. This is presented as an alternative to ad-hoc figures+equations contained in most DL papers.

**Audience:**

Yes

**Broader Impact Concerns:**

No concerns.

**Claims And Evidence:**

Yes

**Requested Changes:**

My main requested change is the hand-holding I suggested above. I know I come off as mostly negative in my review but I believe in the work. I do think my inability to understand most of the paper says something about it; either this paper is meant to appeal to mathematicians analyzing deep nets (in which case I am the wrong reviewer & audience), or this paper is overestimating the reader's familiarity with some concepts and thus giving a presentation that is confusing to most.

I'm of course assuming here that I'm a representative ML researcher.

**Strengths And Weaknesses:**

Starting with strengths; I really appreciate the work that was put into this paper. It's no easy task to think of a (visual!) language that properly defines an extremely wide set of algorithms. I think this paper has value on that sole basis. It's also an interesting (I'm not sure what to call it?) socio-technical contribution, rather than the usual empirical paper.

That being said, I'm honestly not sure what to make of what the authors propose. This is perhaps on a more personal & opinion-level, but in some sense this is where the paper is trying to appeal to the reader. The technical contribution of the paper is obvious, but the paper is claiming to try to do more than that; it's suggesting ML as a field changes its paper-writing culture. I'm having difficulty with the latter.

I have to say when I first scrolled through the paper and happened upon Figures 15 & 16 I was a bit skeptical; too high information density can be just as bad as too low density.

Another question I ask myself, of course inspired by [XKCD 927](https://xkcd.com/927/), is whether the proposed approach is constraining or liberating. Does it enhance our ability to think of new architectures and communicate them? Or does it narrow our thinking [1] to only algorithms that can be expressed (beautifully [2]?) within the language?

[1] The Hardware Lottery, Sarah Hooker, 2020
[2] Lost in Math: How Beauty Leads Physics Astray, Sabine Hossenfelder, 2018

Let me be more concrete.

The basis of what is meant to be widely adopted is already non-trivial. Students come out of a typical CS undergrad knowing linear algebra; as far as I know, category theory, not so much. I certainly only have a superficial understanding of it after 15+ years of ML. Perhaps I'm not the right reviewer for this, but in some other sense, am I not the perfect audience? I suspect the authors want to target _me_.

After reading the paper, I must admit I'm still fairly confused and I don't know how to parse the proposed diagrams. I think part of it may be my lack of familiarity with the basics, but I think the presentation could be significantly different. To be honest, if this work is meant to appeal to a wide audience, there should be way more hand holding. Two suggestions:
- associate each diagram with its equivalent in numpy/torch/some current standard
- associate each diagram with the corresponding linear-algebraic operation (broadcasts and shapes can typically be expressed with proper indexing; this feels feasible)

At least for the first few diagrams, this would be helpful for the more code/linear algebra minded to understand this work. I'm also still unsure what dashed lines represent. Dotted boxes similarly seem to have an important semantic meaning, from what I've gathered elementwise application of a function, but that doesn't seem to work everywhere.

There is one other major downside to the proposed approach: software and science, especially an empirical science like ML, I think is aptly described as a stack of abstractions. The neat thing about that is that if I know what a self-attention block is, and I suspect the reader knows what it is (with an appropriate reference lack-thereof) then I can simply make a figure using self-attention as a building block. We take for granted now that a "neural net layer" is well represented by a matrix multiply + a non-linearity, so much so we don't even need to mention it anymore in papers; this wasn't the case in the 80s. This is great, and leaves more space for more interesting stuff. If we had to redefine everything from scratch in every paper, we'd have enormous papers. I worry that as architectures grow more complex or differ from the usual thinking, neural circuit diagrams would grow to unwieldy sizes. Already, Figure 19 is a whole page; some papers use transformer encoder-decoders as just a block in their overall architecture.


Some other comments:

> The reader must understand a specific programming framework

Presumably here the reader must understand a specific diagram language.

>  Improved communication of architectures, therefore, will offer clear scientific value.

I don't really disagree of course, but I feel like this paragraph presents a false dichotomy. Deep learning's impact on the world was catalyzed by the existence of well written open source software such as theano, pytorch, gym, and so on, as well as well written open source implementations of various architectures that could easily reused. At the end of the day, it's unlikely there will be such a thing as a pure description of an ML algorithm; should we also list the random number generator algorithms used to initialize the model as part of an architecture? A static artifact that we can safely rely on is the code that produced a paper and the corresponding package versions used (or better, some container like Docker cementing the environment in which some code ran); these are becoming more popular and expected from papers.

---

> ### Author Response · Authors · 2023-09-11
> **Thank you for your review!**
>
> Thank you for your review! We appreciate the advice and would happily modify the paper to appeal more to you.
>
> First, [the previous version](https://openreview.net/notes/edits/attachment?id=S721nVCCrj&name=pdf) of this paper is more accessible. If you respond with a quick message saying you prefer its presentation of Neural Circuit Diagrams, we will be happy to use something more like it instead of the current, category-theory-heavy approach. Of course, we will keep the other aspects of the new version, including a more thorough introduction and the coverage of many more cases.
>
> > associate each diagram with its equivalent in numpy/torch/some current standard
>
> Done. Find attached a Jupyter notebook that implements many of the diagrams as PyTorch code.
>
> > associate each diagram with the corresponding linear-algebraic operation (broadcasts and shapes can typically be expressed with proper indexing; this feels feasible).
>
> Shockingly, linear algebra notation can't express broadcasting in a non-obvious way (see: [Named Tensor Notation](https://openreview.net/forum?id=hVT7SHlilx)). Indexes proliferate far too quickly to express large algorithms.
>
> ## Technical Depth v Accessibility
> The most difficult aspect of this paper is getting the right balance between technical depth and accessibility. The technical depth can get immense, these diagrams are extremely robust, and can be generalized and abstracted while maintaining clear meaning. For example, it is possible to make diagrams probabilistic or to incorporate quantum algorithms.
>
> This version's very dense technical presentation was meant to address the concerns of Reviewer 1 about having a sufficient technical contribution. Reviewer 2's favorable assessment of that original version was received a month later, when we had already revised the paper. However, it may have made the paper too dense. [The previous version](https://openreview.net/notes/edits/attachment?id=S721nVCCrj&name=pdf)  is more accessible, and we would be happy to replace parts of the current paper with that version. If needed, we could remove all the category theory.
>
> ## Necessity of Diagrams
> We feel it is necessary to change the paper-writing culture of ML, and that TMLR is the correct forum to suggest such changes. A number of the [scope guidelines](https://jmlr.org/tmlr/editorial-policies.html) apply to our work, with "new approaches for analysis, visualization, and understanding of artificial or biological learning systems" being the most relevant. It is similar in ambition to [Named Tensor Notation](https://openreview.net/forum?id=hVT7SHlilx), a previous TMLR paper that took a non-graphical approach.
>
> We feel that the graphical approach is liberating. It lets algorithms be conceptualized independent of an underlying programming language, and lets them be viewed and understood at a glance. Section 3.5 argues this the best, where vision transformers naturally emerge when looking at neural circuit diagrams. This type of conceptualization is more difficult in a non-graphical approach, especially when the changing size of data is hard to keep track of.
>
> > Presumably here the reader must understand a specific diagram language.
>
> Concerning this comment and XKCD 927, currently we have one diagramming language, effectively, *per paper*. Given the importance of visualization to humans, people will want to visualize whatever they are working on. Without a formal diagramming language, ad-hoc diagrams will proliferate instead.
>
> Furthermore, these diagrams are hardly introducing a new standard. They are closely related to monoidal string diagrams, which have seen consistent success in a [variety of fields](https://arxiv.org/abs/0903.0340). A famous example are Feynman diagrams. The intersection of monoidal string diagrams and having both Cartesian and tensor products gives Neural Circuit Diagrams. Therefore, other formal standards are unlikely to deviate much from what we present.
>
> ## Abstractions
> Abstracting across different scales is a distinct advantage of our diagrams. We have added two new paragraphs, paragraphs 2 and 3 of Section 1.3, that address this in greater detail. We fully intend diagrams to vary based on the target audience and to embrace a generous amount of abstraction. Diagrams accept both vertical and horizontal composition, meaning regions of diagrams can effectively be fenced off and reduced to well-defined blocks that can be used later. Some of our implementations show this.

---

> ### Author Response · Authors · 2023-09-24
> **Updates on latest revision**
>
> In the latest revision, we have restructured the introduction to cover the philosophy of our approach, addressing key questions you raised about how diagrams should be used regarding levels of abstraction and whether contributing a diagrammatic standard is worthwhile.
>
> Additionally, we have redone Section 2, introducing neural circuit diagrams, to be more accessible by removing the category theory. The technical foundations are still there without using mathematics that will dissuade adoption. Furthermore, we have added additional technical depth in Section 3.6, covering differentiation, and showing the utility of neural circuit diagrams for analysis by deep learning researchers.
>
> The paper is now more focused and specialized for an audience of deep learning researchers. Diagrams and concepts are introduced with gradually increasing complexity while covering the technical foundations. The accompanying code was a great suggestion and helps explain the diagrams to readers familiar with PyTorch and wondering how diagrams relate to implementation.
>
> Overall, we feel the paper is in a strong position to encourage adoption by deep learning researchers while leaving open future research into generalizing and formalizing diagrams with category theory.

---

> > ### Comment · Reviewer_wTkA · 2023-10-16
> >
> > Thanks! I actually went back and read the previous version of the paper, and I must admit that I think I liked it better, but the latest revision is great too. I will recommend acceptance.

---

> > > ### Author Response · Authors · 2023-10-19
> > > **Thank you for your support**
> > >
> > > We appreciate your support and have found your reviews valuable while working on the paper. We have just implemented a series of minor changes from the suggestions of 82Vd, and we are very happy with the current state of the paper.

---

### Decision · Action_Editor_2ssq · 2023-11-25

**Recommendation:** Accept as is

**Comment:**

The paper itself is unconventional, but I think that is precisely the sort of thing TMLR is designed for. It underwent multiple major revisions through the review process, first adding more technical depth and then dialing back and making it more accessible. The current version strikes a reasonable balance between the two. The addition of code examples and extension to many contemporary architectures greatly clarifies things and makes it more compelling for readers in the field.

The paper has already gone through so much editing that I hesitate to condition acceptance on further changes, since I don't think it will dramatically affect the paper. However, I would encourage the authors to consider Reviewer 82Vd's advice about adding further explanation for transpose and convolution operations.

**Audience:**

One question that appeared among several reviewers is whether this would see broad adoption within the community. I would tend to agree with them that I don't think that this is necessarily the case, but it is a valiant attempt to bring a common visualization language to the field and it would be wonderful if some form of standardization did get adopted. In that regard, I do believe that this will get more people thinking about visualization and perhaps some of the ideas here will be adopted in the community.

**Claims And Evidence:**

Yes, while the paper doesn't make any empirical claims, it does develop a new form of visualization for neural network architectures and visualizes a number of contemporary architectures under this diagramming framework.